# SETTING UP FOR FAILURE: AUTOMATIC DISCOVERY OF THE NEURAL MECHANISMS OF COGNITIVE ERRORS

**Puria Radmard**[*]    **Paul M. Bays**[†]    **Máté Lengyel**[*,‡]

[*]Department of Engineering, University of Cambridge
[†]Department of Psychology, University of Cambridge
[‡]Department of Cognitive Science, Central European University

`{pr450,pmb20,ml468}@cam.ac.uk`

## ABSTRACT

Discovering the neural mechanisms underpinning cognition is one of the grand challenges of neuroscience. However, previous approaches for building models of recurrent neural network (RNN) dynamics that explain behaviour required iterative refinement of architectures and/or optimisation objectives, resulting in a piecemeal, and mostly heuristic, human-in-the-loop process. Here, we offer an alternative approach that automates the discovery of viable RNN mechanisms by explicitly training RNNs to reproduce behaviour, including the same characteristic errors and suboptimalities, that humans and animals produce in a cognitive task. Achieving this required two main innovations. First, as the amount of behavioural data that can be collected in experiments is often too limited to train RNNs, we use a non-parametric generative model of behavioural responses to produce surrogate data for training RNNs. Second, to capture all relevant statistical aspects of the data, rather than a limited number of hand-picked low-order moments as in previous moment-matching-based approaches, we developed a novel diffusion model-based approach for training RNNs. To showcase the potential of our approach, we chose a visual working memory task as our test-bed, as behaviour in this task is well known to produce response distributions that are patently multimodal (due to so-called *swap errors*). The resulting network dynamics correctly predicted previously reported qualitative features of neural data recorded in macaques. Importantly, these results were not possible to obtain with more traditional approaches, i.e., when only a limited set of behavioural signatures (rather than the full richness of behavioural response distributions) were fitted, or when RNNs were trained for task optimality (instead of reproducing behaviour). Our approach also yields novel predictions about the mechanism of swap errors, which can be readily tested in experiments. These results suggest that fitting RNNs to rich patterns of behaviour provides a powerful way to automatically discover the neural network dynamics supporting important cognitive functions.

## 1 INTRODUCTION

An important goal for computational neuroscience is to use behavioural data to generate viable and testable hypotheses about the neural network mechanisms that underpin cognition. To achieve this goal, the following requirements need to be met: (1) hypotheses should be formulated as recurrent neural networks (RNNs) so that dynamical neural mechanisms can be studied (2) models should be quantitatively fit to behaviour, such that they capture as many statistical properties of the data as possible, as important cognitive mechanisms have been shown to only reveal themselves in detailed patterns of response variability (rather than in averages); (3) model fitting should be automated, avoiding subjective, piecemeal iterative model construction, so neural network mechanisms can be accelerated at scale.

Previous approaches have fallen short of these desiderata. For example, cognitive models – including drift-diffusion (Resulaj et al., 2009; Pardo-Vazquez et al., 2019), Bayesian (Heald et al., 2021), symbolic (Castro et al., 2025), and large language model-based models (Binz et al., 2025) – have

been successfully fit to detailed behaviour with impressive predictive power, but their architectures remained too abstract and divorced from RNNs to be useful as hypotheses about neural network dynamics (thus failing Requirement 1). Neural population coding models have suggested causal connections between neural response properties and particular patterns in behaviour (Matthey et al., 2015; Schneegans and Bays, 2017; McMaster et al., 2022), but remained mute about the neural network dynamics that give rise to the hypothesised neural responses in the first place (failing Requirement 1), have been rarely fit to behaviour quantitatively (failing Requirement 2), and their construction was not automated (failing Requirement 3). RNN models – either with hand-crafted (Edin et al., 2009; Bouchacourt and Buschman, 2019), or with task-optimized architectures and parameters (Mante et al., 2013; Stroud et al., 2023) – have been highly successful at suggesting testable hypotheses about neural dynamics underlying cognitive performance, but they rarely capture behaviour beyond generically competent performance, let alone being quantitatively fit to detailed behavioural data (failing Requirements 2, and for hand-crafted networks also Requirement 3). In some cases, non-trivial aspects of behaviour were successfully captured by such models, but at the expense of including purpose-built design choices or ablations (Xie et al., 2023), thus making these models essentially hand-crafted in this sense (failing Requirement 3).

In this work, we introduce an approach that satisfies all three requirements. We extend automated neural mechanism discovery by directly training biologically plausible RNNs to exhibit behavioural suboptimalities seen in experimental data. We tackle two major challenges for doing so. First, RNNs are data hungry, and behavioural data is scarce. To make training feasible, we generate synthetic data using a descriptive generative model, and use this to train our RNN (Figure 1). The second challenge

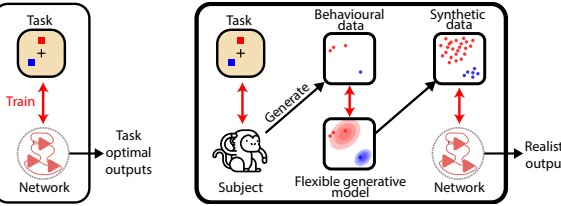

Figure 1: Left: typical procedure involves training an RNN to perform optimally in a task, without relating to real behaviour. Right: our novel method can replicate subject behaviour, by training on surrogate training data.

is defining a suitable training objective for the RNN so that they can generate complex (for example, multimodal (Bays et al., 2009) or skewed (Fritsche et al., 2020; Resulaj et al., 2009)) continuous response distributions often found in experiments. Previously used methods for training RNNs are not appropriate for this. Task optimization of RNNs typically used simple optimisation criteria, such as mean squared error or cross entropy (Yang et al., 2019), which encourage unimodal, typically normal response distributions. Even when training RNNs explicitly to generate specific distributions of responses, moment- or score matching-based criteria were used (Echeveste et al., 2020; Chen et al., 2023), which also do not scale well to capturing complex distributions (and require arbitrary choices as to which moments or statistics of the target distribution are important). Instead, by framing elicitation of a behavioural output as generating samples in Euclidean space, we train the RNN using a training criterion inspired by diffusion models (Ho et al., 2020), a state-of-the-art class of generative neural models for complex, continuous distributions. We adapt the training procedure used for diffusion models to work within an RNN, harnessing their flexibility in sampling from complex distributions. This enables us to flexibly fit to continuous response data, unlike prior methods of fitting directly to behaviour (Ji-An et al., 2025), which are limited to categorical distributions (§5).

We demonstrate the power of our approach for automatic mechanism discovery on one of the most studied cognitive paradigms whose neural underpinning is still poorly understood: visual working memory (VWM). While there are many neural circuit models of working memory, they typically do not address the more challenging (and ecologically more relevant) situation when multiple pieces of information ('items') need to be maintained in memory Zhang (1996). Even RNNs that do perform multi-item VWM tasks fail to capture the characteristic patterns of behaviour that are specific to the multi-item situation (Yang et al., 2019; Driscoll et al., 2022), or only capture some select behavioural biases by using hand-crafted networks with purpose-built architectural motifs with limited biological plausibility (Bouchacourt and Buschman, 2019). In particular, so-called *swap errors* are a major source of errors that arise when participants recall the wrong item from their working memory, instead of the item that has been cued – and thereby create strongly multimodal response distributions (Bays et al., 2009). Current RNN models do not capture swap errors, and the resulting multimodal response distributions, at all. Conversely, there exist population coding models that do capture swap errors, but do not make predictions about neural circuit mechanisms; Schneegans and Bays (2017)). We address this gap by showing that our approach produces neural circuits that generate rich, realistic response

distributions by directly training networks to perform swap errors. We use our trained networks to generate hypotheses about the biological implementation of observer models from the psychophysics literature (Schneegans and Bays (2016); McMaster et al. (2022); §4), previously inaccessible to RNNs trained for task performance.

## 2 BACKGROUND

**RNNs in neuroscience**   Neural models of biologically plausible cortical circuits are described by their continuous-time "leaky" dynamics:

$$\Lambda \dot{\boldsymbol{r}} = -\boldsymbol{r} + \mathcal{F}(\boldsymbol{r}, \boldsymbol{s}, \boldsymbol{\eta}; \theta_r) \tag{1}$$

where $\boldsymbol{r}$ is the vector of neural firing rates (hidden RNN activations), $\boldsymbol{s}$ is the input from an external population (e.g. sensory input into the population), $\boldsymbol{\eta}$ is the population's biological process noise, and $\Lambda$ is the diagonal matrix of neurons' membrane time constants (typically constant across cells, i.e. $\Lambda = \lambda I$). The non-linear computation $\mathcal{F}$, parameterised by $\theta_r$, models the recurrent communication between and information integration within neurons. The network output is generically $\boldsymbol{x} = \mathcal{G}(\boldsymbol{r}; \theta_o)$, often interpreted as a behavioural response. The exact form of $\mathcal{F}$ and $\mathcal{G}$ vary across studies; in this work, we use a dendritic tree architecture for each neuron (Appendix B; Lyo and Savin (2024)), and interpret the output of the network - a simple linear projection of the firing rates - as a colour estimate on the colour circle. More details are provided in §3, and discussion of our choice of architecture is provided in Appendix B - namely, we chose this architecture to be sufficiently expressive for the task posed, while also maintaining a biologically plausible connectivity structure.

Previous attempts to train RNNs that generate samples typically train RNNs on task-optimality or moment-matching criteria (Echeveste et al., 2020). This is insufficient for us - as outlined in §1, we are interested in reproducing behaviour that takes on a distinctly multimodal distribution. To do so, we employ methods from the diffusion model literature. We briefly introduce diffusion models here, but defer many details to previous foundational work (Ho et al., 2020).

**Denoising diffusion probabilistic models (DDPMs)** DDPMs are a class of generative models that involve a neural network learning to undo a fixed noising process applied to data samples. During training, some $\boldsymbol{x}_T$ is drawn from data, and is iteratively corrupted:

$$q(\boldsymbol{x}_{\tau-1}|\boldsymbol{x}_\tau) = \mathcal{N}(\boldsymbol{x}_\tau; \sqrt{1-\beta_\tau}\boldsymbol{x}_\tau, \beta_\tau I) \tag{2}$$

which admits a closed form posterior $q(\boldsymbol{x}_{\tau+1}|\boldsymbol{x}_\tau, \boldsymbol{x}_T)$ with mean $\boldsymbol{\mu}_q(\boldsymbol{x}_\tau, \boldsymbol{x}_T)$. The DDPM describes a non-stationary transition kernel $p_\phi(\boldsymbol{x}_{\tau+1}|\boldsymbol{x}_\tau, \tau)$, and is trained by maximising a hierarchical evidence lower bound (ELBO) of the one-step denoising of the corrupted data. By parameterising the transition kernel by its mean $\hat{\boldsymbol{\mu}}_\phi(\boldsymbol{x}_\tau, \tau)$ only, then adding isotropic Gaussian noise with the same variance as the noising posterior, this ELBO can be expressed as a sequence of square distances, rather than KL divergences. We equate timesteps within the trial to timesteps in the DDPM denoising process, and use this objective to train the RNN to generate realistic samples from the response distribution. Training RNN-like diffusion models to perform cognitive tasks further requires a form of teacher forcing (Appendix D).

**Synthetic data**   Neural networks are data-hungry. This is not an obstacle training for task-optimality – task variables such as stimulus features and target responses can be generated procedurally – but it is when training directly on behaviour – producing human or animal behavioural datasets is prohibitively costly at the scales required for training networks. Therefore, we generate synthetic behavioural data to plug this gap. Many of the models listed in §1 which accurately describe behaviour using neural population codes (Schneegans and Bays, 2017), Bayesian cognitive (Heald et al., 2021) and descriptive (Bruijns et al., 2023) models, and/or large language models (Binz et al., 2025) can, by the same token, be used to faithfully generate synthetic behavioural data. These synthetic datasets can be used to train our dynamical neural model; sufficient flexibility in the generative model allows synthetic data that captures desired statistical facets and suboptimalities of behaviour. Many of these models, or their relatives, have also been used to fit behavioural data from working memory (WM) tasks, with varying degrees of satisfaction of our three requirements. For this reason, and because of the sophisticated response distributions elicited from even simple WM tasks, we choose this as the testbed for our method. In §5, we will return to potential extensions to other aspects of WM, and other cognitive tasks in general.

**Working memory (WM) and swap errors**    While not consistently distinguished from short-term memory (Aben et al., 2012), WM is generally defined as the temporary maintenance, manipulation, and retrieval of sensory information over an order of seconds, for executive functions. A dominant experimental paradigm in investigating visual working memory (VWM) is the *delayed estimation* task, in which a human or animal subject is presented with sensory information and asked, after a short delay during which the stimulus is withdrawn, to reproduce some information about it. Figure 2 shows an example of a *multiitem* delay estimation task, where the subject is presented with multiple (here, 2) items, then presented with a *probe dimension* (here, location) feature value, called a *cue*, and asked to reproduce the corresponding *report dimension* (here, colour) feature value of the cued item. From here, we will interchangeably use 'location' and 'probe feature', and 'colour' and 'report feature' for ease of language, however the features used for each role is open for experimental design. Most responses fall near the cued item's report feature value, and the distribution of estimation errors along the report feature (here, the colour circle) provides key evidence for measuring the capacity of VWM (Ma et al., 2014).

However, there is also a central tendency in responses towards the colour of *uncued items*, a.k.a. *distractors* (Bays et al., 2009; Gorgoraptis et al., 2011). This indicates the presence of *swap errors*, in which subjects mistakenly recall the colour of a distractor. Behavioural and neural evidence largely implicates *misselection* of the correct item from memory at cueing time, rather than a misbinding of feature values during memory encoding or storage – swap errors are a failure to extract the correct item from memory, rather than a misencoding into memory (but see Radmard et al. (2025)) Macaque neural signatures reflects this misselection hypothesis Alleman et al. (2024). The key evidence for this is that swap errors are more likely to happen when a distractor's probe feature value is close in feature space (here, physical space) to the cued one (Emrich and Ferber, 2012; Schneegans and Bays, 2017; McMaster et al., 2022). Describing the expected distribution of responses, including swap errors, requires a multimodal distribution, with modes placed over both target and uncued colours.

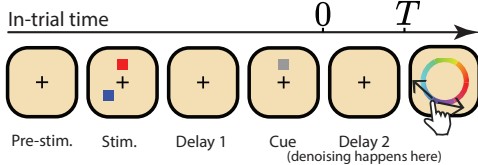

Figure 2: Two-item delayed estimation task - minimal VWM task for swap errors.

**Bayesian Non-parametric model of Swap errors (BNS)**    We use BNS (Radmard et al., 2025) to generate synthetic multiitem delayed estimation data. This model predicts the likelihood of a swap error as a function of the distance of the distractor from the cued stimulus in each feature dimension (Appendix A). In the present work, we do not directly fit BNS to a real behavioural dataset. Instead, we opt to hand-tune the generative model to capture archetypal dependencies of swap errors. Specifically, we train RNNs with synthetic data generated by instantiations of BNS that either i) predicts no swap errors, ii) predicts the same frequency of swap errors for any distractor, and iii) predicts swap errors more frequently when its probe feature is more similar to that of the cued item, as per the extensive experimental evidence. In §3, we show that RNN representations resemble the real macaque neural geometry only in the third case, when maximal realism is captured in the training data, even in non-swap trials. We also fit BNS to the behavioural output of the trained RNNs to quantify its success in reproducing the target dependence on probe difference (§4). Again, we emphasise that the use of VWM and BNS is one instantiation of our method, and that its ethos of fitting directly to suboptimal behaviour extends beyond this domain, which we discuss briefly in §5.

## 3    METHODS

**Network architecture**    Our RNN models a population of $n$ densely connected cortical pyramidal neurons[1]. Each neuron integrates inputs from a sensory population and all other neurons in the population via a dendritic tree (Appendix B; Lyo and Savin (2024)). Overall, $\mathcal{F}$ in our network dynamics (equation 1) is the somatic integration of the final, most proximal, layer of dendritic nodes with additive Gaussian noise (Appendices B, G). We train the RNN to perform the multiitem delayed estimation task (§2) with 2 items, providing the minimum viable task in which swap errors could arise. The network is first provided with two stimuli, each parameterised by two features, dubbed probe and report (visualised as location and colour), both on a circle. For *index-cued* networks, only the stimuli colours are provided as individual ordered scalars in $[-\pi, +\pi)$, and the cue as an

---

[1]Constants used in defining the architecture/training processes are declared in Appendix G.

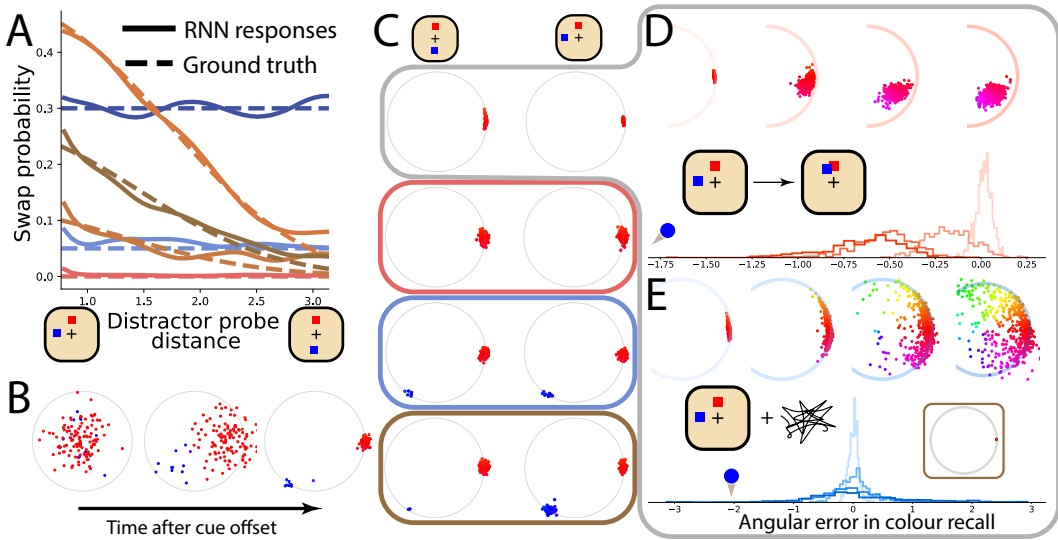

Figure 3: DDPM-style trained RNNs accurately captures swap errors in training data, unlike ablated task-optimal networks. **A** dotted lines are the target swap rate used to generate the synthetic training data for various RNNs, and solid lines are average swap rates inferred from fitting BNS to RNN-generated behaviour after they are trained on this synthetic data (Appendix A; Radmard et al. (2025)). RNNs achieve fair success in replicating training data for most trials in both probe distance-dependent and -independent cases. Corresponding sample sets in $\mathbb{R}^2$ are shown in **C**, where border colours match the line colours here. **B** Trajectory snapshots at different points in the second delay period, during which the network is trained to denoise behaviour. Trials are coloured by their argument at the end of denoising, which is interpreted as the colour estimation made by the network. Final samples successfully sample from their multimodal target distribution (Appendix A). **C** Typical set of final timestep samples for trials with close (left) and far (right) probe feature values. In all cases the red stimulus is cued. Borders indicate generative RNNs, referencing lines in **A**. Top row (grey border) indicates task-optimal network. **D,E** Two typical ablations that may be applied to task optimal networks to indcue swap errors: decreasing probe distance beyond the minimum margin between items seen during training to induce confusion, and increasing process noise variance beyond training conditions to increase misselection likelihood. Swap errors would be evidenced by a second mode at the colour of the distractor, indicated by a blue marker. This does not arise in either case, and further attempts would require subjective ablation design. **E** inset: Conversely, removing process noise for a network trained to perform swap errors recapitulates optimal behaviour.

index. This provides a direct analogy to the tasks performed by macaques (Panichello and Buschman, 2021; Alleman et al., 2024), to which we make comparison (§4). Locations are not provided to these networks, and colours are ordered. For *feature-cued* networks, the stimuli are provided to the network as the conjunctions of locations and colours. These are fed to the network as activations of a palimpsest conjunctively tuned sensory neural population (Appendix B; Matthey et al. (2015); Schneegans and Bays (2017)). Importantly, in the latter case, items are order invariant - the network must store the two conjunctions between the two features. This allows us to extend to predictions about the representation of a variable probe feature. The network's firing rates vector is projected to the 2D plane with a *fixed, orthonormal* output projection matrix $\boldsymbol{x} = \mathcal{G}(\boldsymbol{r}) = W_x\boldsymbol{r}, W_x \in \mathbb{R}^{2\times n}$ (i.e. $W_x^\intercal W_x = I_n$). Network activity is therefore split into two fixed orthogonal subspaces: the *behavioural subspace* (the network output) $\boldsymbol{x} = W_x\boldsymbol{r}$, and the *behavioural nullspace* $\boldsymbol{m} = W_x^\perp\boldsymbol{r}$. We interpret the argument of this output at the final timestep of the trial, $\arctan\left([W_x\boldsymbol{r}_T]_1/[W_x\boldsymbol{r}_T]_2\right)$, as the colour recalled by the network on that trial. To train for task performance, a mean squared error (MSE) loss on the discrepancy between this 2D estimate and the report feature of the cued item embedded on the circle in $\mathbb{R}^2$ suffices (Appendices A, D).

**Training** Instead of training for task optimality, we adapt the training procedure used for generative diffusion models (§2; Ho et al. (2020); Lyo and Savin (2024)) to train for behavioural realism, accounting for the presence of swap errors. We apply the DDPM training objective only to $\boldsymbol{x}$, not the full activity vectofr $\boldsymbol{r}$, equating timesteps after cue offset to timesteps in the DDPM denoising process. Besides the use of leaky dynamics (see above), this is a key difference to previous integrations of

diffusion models with RNNs in cortical modeling (Lyo and Savin, 2024). Additionally, the DDPM-style criterion is only applied to the $T$ timesteps following cue offset (Figures 2, 3B) – not repeatedly throughout the full duration of the trial – and the target distribution is trial-specific.

For each task trial, the target distribution of responses is defined by a mixture of 2 Gaussians in $\mathbb{R}^2$, with mode means determined by colours of stimuli (on the circle), and weights determined by BNS prediction (Appendix D). This final behavioural sample is then iteratively noised (equation 2) to produce the target trajectory for the second delay. The network is trained to undo this process at each timestep of the second delay, with the transition kernel mean provided by discretising the continuous noiseless RNN dynamics (equation 1) projected to the behavioural subspace:

$$\hat{\boldsymbol{\mu}}_{\theta_r}(\boldsymbol{r}_t, t) = W_x \left[ \left( 1 - \frac{\mathrm{d}t}{\lambda} \right) \boldsymbol{r}_t + \frac{\mathrm{d}t}{\lambda} \mathcal{F}(\boldsymbol{r}_t, \boldsymbol{s}_t; \theta_r) \right] \tag{3}$$

Note that because diffusion only occurs during the second delay, the sensory term $\boldsymbol{s}_t$ is empty (Appendix B), and information about the full set of stimuli, and hence the target distribution, is contained in the behavioural nullspace activity $\boldsymbol{m}_t$. This dependence on previous timesteps means training requires simulation of the full trial trajectory in sequence. Because valid application of the DDPM criterion (§2) requires training examples of noised data ($\boldsymbol{x}_\tau$) to be drawn from the marginal distribution of the noising process, i.e. $q(\boldsymbol{x}_\tau | \boldsymbol{x}_T)$, we employ teacher-forcing during training to ensure that the distribution of trajectories of $\boldsymbol{x}_\tau$ stay in distribution. The training algorithm is summarised in algorithm 1, with full details provided in Appendix D.

---

**Algorithm 1** Summary of training with teacher-forcing - full algorithm in Appendix D.

---

**repeat**
    Generate stimulus set and select cued item
    Using BNS, generate a sample $\boldsymbol{x}_T^*$ from the desired behavioural distribution in $\mathbb{R}^2$, and apply the noising process (equation 2) to generate noised data $\boldsymbol{x}_{0:T-1}^*$
    Initialise network activity from $\mathcal{N}(0, \sigma_r^2 I)$
    Run network dynamics for the prestimulus, stimulus, delay, and cue periods (Figure 2) with noisy discretised leaky dynamics (equation 3 without projection), finally arriving at network activity at cue offset $\boldsymbol{r}_0$
    **for** $t = 0$ **to** $T$ (i.e. during second delay) **do**
        Apply teacher-forcing with $\boldsymbol{x}_t^*$ (Appendix D)
        Calculate mean transition kernel $\hat{\boldsymbol{\mu}}_{\theta_r}(\boldsymbol{r}_t, t)$ (equation 3)
        Run dynamics one step (equation 3) and add scheduled noise (Appendix G)
    **end for**
    Train on regularised DDPM criterion (Appendix D)
**until** converged

---

## 4 RESULTS

**Swap dependence reproduction** We start by summarising the success in reproducing a variety of desired behaviours, characterised by the target swap function. Figure 3A shows the ability of the RNNs to replicate the dependence of swap errors on distractor distance used to generate their synthetic training data, for a variety forms of dependence (Appendix A). Here, dotted lines show the ground truth swapping rates as a function of probe distance, and solid lines show feature-cued networks' abilities to capture them. This is inferred by fitting BNS to the estimates generated by the trained RNN. Disparities in low swap probability regions is likely due to the compounded lack of training samples for both the RNN and BNS in low probability modes. Example sample sets, coloured by the associated estimate, are shown for some of these in Figure 3C, with the multimodality achievable with our method contrasting the unimodal distribution of estimates produced by task-optimised networks (top row). Figure 3D and E summarise naïve attempts to induce this multimodality after training for task-optimality, which one might expect researchers to attempt. Failure to do so would typically set off an iterative and targetted sequence of ablations based on heuristics/prior assumptions on the origin of swap errors. We achieve swap errors by directly training for them, and find that the resulting neural signatures underlying these errors match those found in biology, as we show below.

**Neural representations of memoranda** We now compare the neural representations of stimuli during each delay to those of macacque lateral prefrontal cortex (LPFC) during a similar task

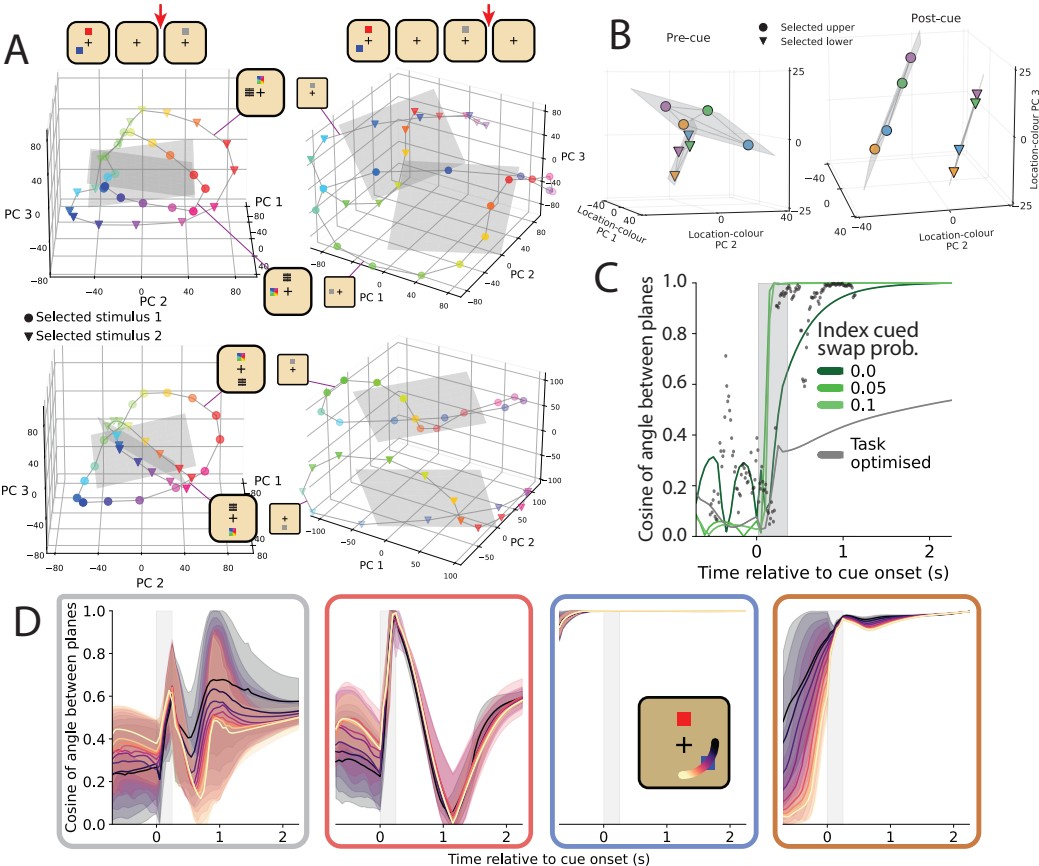

Figure 4: Behaviourally realistic, but not task optimal, networks capture neural signatures of VWM. **A** $m_T$ averaged over many accurate trials with distractor report feature varying, for trials with close (top) and far (bottom) stimulus distance, before (left, planes misaligned) and after (right, planes aligned) cue exposure. Colours represent report feature of cued (and accurately recalled) stimulus. Representations drawn from network in **E** (right), which best matches neural data (see below). **B** This qualitatively matches equivalent representations in macaque cortex - in this case there are only two possible cue values. We find a similar pre- and post-cue geometry for our index-cued networks, which are the closest analogue to the real task. **C** The real data (dots) further shows that this planar alignment (quantified as cosine similarity between normals) increases before and during cue exposure (grey stripe), and remains high throughout the second delay until time of recall. This is matched for our index cued networks when trained with a data-matched proportion of swap errors (5-10%). Even training with our DDPM-based method with a no-swap target distribution does not achieve this rapid increase during the cue period. A longer second delay period was chosen to encompass all experimental conditions in the original study. **D** Plane alignment over time for the feature-cued networks. Axes borders indicate the model used, as in Figure 3 . Importantly, *only the model with distance-dependent swap probability* matches the description of the biological data like its index-cued counterparts. This is quantified in Appendix C. Error bars show mean ± std across all different spatial configurations with the denoted stimulus location distance (see inset; purple is closest distractor, yellow is furthest). **B C** generated with data from Panichello and Buschman (2021)

(Panichello and Buschman, 2021). Here, two macacque monkeys performed a version of the multiitem delayed estimation task with fixed stimulus locations - one coloured square in the top and one in the bottom half of their field of vision. A binary visual cue at the point of fixation indicated which item is to be recalled. This task is most analogous to the *index-cued* version of our networks. As such, following subsections will begin by reproducing biological phenomena with these networks, before going on to make predictions about the role of the probe feature with our *feature-cued* networks. For the former, we match behavioural statistics; for the latter, we construct archetypal swap dependencies as the target distribution, to form predictions about neural representations.

Their dataset of two monkeys for this retrospectively cued task (4,769/3,943 completed retro trials over 13/10 sessions) revealed: i) a swap error rate of roughly 10% when pooling the data for the two monkeys, and ii) a consistent low dimensional representation of the stimuli in memory before and after the onset of the cue during accurate trials (Figure 4B). At both points in accurate trials (i.e. no swap error, and low angular error), the mean population response of LPFC associated with the colour of each item (upper and lower) was found by averaging neural activity across trials with that colour in that position, factoring out the colour of the distractor. These representations formed a 2D cycle when averaged across trials, reflecting the cyclicality of the behavioural responses provided by the monkey on the colour circle. Prior to cue onset, the representation of each item occupied perpendicular 2D planes (Figure 4B [left]) and after cue onset, these planes reoriented to become parallel (Figure 4B [right]), with the cosine of the angle between the planes increasing to a maximum during the cue remaining high (i.e., parallel) during the second delay (Figure 4C).

We compare these findings to representations in the behavioural nullspace activity $m$ of our RNN, for index- and feature- cued networks, as well as task-optimised networks. For index-cued networks (Figure 4C), which most closely analogise the real task, optimisation for task performance does not capture the full alignment of the stimulus representations during and after cue presentation. The sharp and sustained increase and alignment is only captured when using our DDPM-style training *and* including swap errors in the trial-wise target distributions.

In the feature-cued case, where a continuous location is used to load and cue items from memory, this pattern is only captured when the network is trained with a location-dependent swapping rate (Figure 4D [right]), and not with a flat, probe feature-independent rate of swap errors. As motivated above, while we matched behavioural statistics for the index-cued task, which bears direct resemblance to the real task provided to monkeys, we modelled the form of the swap dependence off more sophisticated tasks (Emrich and Ferber, 2012; Schneegans and Bays, 2017; McMaster et al., 2022) on arc in this task in order to make predictions about their neural representations. Furthermore, this network bears higher pre-cue representational alignment for distractors with closer probe feature values. This provides a prediction for the neural substrate of delay-time processes leading to memory retrieval error at cueing, previously described in the psychophysics literature as noising or drift in the abstract representations of stimuli along the probe dimension over time (Schneegans and Bays, 2017; McMaster et al., 2022). **Concretely, we predict that trial-by-trial decomposed analysis of the alignment between stimulus representations will reveal that swap errors occur more often when pre-cue representational alignment of colour is higher.** This could be due to probe similarity, as in the case presented, or other factors such as attention before cueing in the fixed item location case.

**Item misselection** The previous subsection made the prediction that physically closer, hence more swap-prone, distractors cause higher report feature representation alignment before cue onset (Figure 4D [right]). We now consider the neural signatures of the retrieval errors at cueing time, which lead to swap errors. For the same macaque task, experimentalists later showed that *item misselection* is the primary cause of swap errors (Figure 5A; Alleman et al. (2024)). The misselection is operationalised as follows: (i) before cue onset, items are stored as conjunctions between their probe (location - or just upper and lower in the experimental case) and report feature (colour) values (Figure 5A [left]); after cue onset, items are stored as conjunctions between their *role* (target vs. distractor) and report feature value (Figure 5A [right]) These are dubbed the *spatial* and *report* subspaces respectively; item 'misselection' is misprojection between these subspaces at cueing time. Swap errors can also arise at cueing time due to *misinterpretation* of the cue variable. Alternatively, report and probe features can be *misbound* prior to cueing. Alleman et al. (2024) find the least evidence for this mechanism, instead asserting that swap errors arise during or after cueing time. We now complement existing evidence of this neural basis of swap errors, which depends on across-trial statistical modelling of neural responses Alleman et al. (2024), with the representational geometry learned by our network.

First, we apply the same linear decoding analysis as Alleman et al. (2024) to the memory subspace neural activity of our index cued network. These networks hold direct analogy to the task analysed by this work, so results can be directly compared. This method is based on training linear decoders on non-swap trials, then projecting neural activity onto the line connecting these *nominal* representations to representations that arise due to *misbinding*, *cue misinterpretation*, or *item misselection*. We defer details of this analysis to Alleman et al. (2024). As with the real macaque data, this analysis suggests that swap errors arise in our network during the second delay, rather than a misbound representations of memoranda Schneegans and Bays (2017); McMaster et al. (2022).

Next, we investigate our probe-cued networks for the existence of *report subspace*, where colours are bound to roles (target or distractor) with locations factored out, and the neural signature of swap errors during delay 2. Figure 5C shows a low-dimensional projection of $m_T$ for a fixed set of locations and one fixed colour forming two parallel rings, coloured by the free stimulus. In the left column, when the fixed colour stimulus is cued, the bottom ring (negative PC3) is made up of the representation of swap trials, which are coloured by the uncued but *mistakenly recalled* colour, and the top ring (positive PC3) is coloured by the distractor colour when the fixed colour item is *accurately recalled*. Vice versa when the fixed colour stimulus is uncued (right column). The normal to these rings, which discriminates accurate and swap trials (Figure 5D [bottom]), is independent of the stimulus locations, only depending on the fixed stimulus colour considered (Figure 5D [top]). **This makes this two ring structure a candidate for the geometry of the postcue report subspace** (Alleman et al., 2024), which binds stimulus colour (around the ring) to its recalled role (choice of ring), and is independent of the original colour-location binding (Figure 5A). We have not considered exactly *how* this report subspace varies across different fixed report stimuli, just that the normals of these rings is shared whenever one of the items has a fixed colour, regardless of the item locations or whether the fixed item was cued (Figure 5C [left]). However, their alignment depends on colour similarity (Appendix E), suggesting an intuitive toroidal formation.

## 5 DISCUSSION

In this work, we trained RNNs to exhibit swap errors during a multi-item delayed estimation VWM task (§3). Its success was contrasted with naïve training methods and ablations applied to task-optimal networks (Figure 3). Capturing swap errors *post hoc* would require a series of directed network modifications, likely driven by leading assumptions or heuristics as to how swap errors arise, whereas our method displays them directly. Then, we showed that this training method captured a more accurate description of neural mechanisms when trained to adhere to the empirical variability around real swap errors (Figure 4). Namely, we trained one network to produce swap errors more frequently when there is higher probe similarity between the cued item and the distractor (Figure 4E [left]), previously theorised to increase the likelihood of the item *misselection* underlying swap errors. Separating trials by probe distance, we provided a prediction for the neural representation of probe similarity before the cue. Finally, we provided a candidate representational geometry for swap errors, in line with previous work on their neural basis.

Our approach bridges the gap between flexibly modelling neural circuits and producing rich descriptions of behaviour, that were previously only achieved separately by task-optimised networks and abstract cognitive models, respectively. We did so by reversing the typical sequence of normative modelling - by directly training networks to reproduce the full distribution of behavioural responses, including suboptimalities, we bypass the need for an iterative, potentially heuristic set of ablations. These principles are most tightly shared by tiny RNNs (Ji-An et al., 2025). However these small networks, like previous approaches fitting to behavioural data (Xue et al., 2024) have only been used for discrete behaviour, e.g. categorical choices, for which simple maximum likelihood fitting to behavioural datasets suffices. Our method provides a novel, principled approach to holistically fitting to all, continuous dimensions of data. The neural predictions we made by reverse engineering behaviour in this way await experimental verification, but demonstrate the value of training networks to reproduce behavioural errors rather than optimising for performance.

One limitation of our approach is its dependence on a generative behavioural model to produce training data, which may not be available for all tasks of interest. However, for many well-studied paradigms such models already exist, and recent advances in descriptive modelling (Binz et al., 2025) suggest that capturing the relevant statistical structure of behaviour, rather than normative or mechanistic correctness, is often sufficient and increasingly achievable.

The present work can be extended in multiple directions. We used this VWM task as a testbed for our novel methodology because of its inherent multimodality, but our method can be extended to any other cognitive task associated with complex behavioural suboptimalities, or other non-trivial response distributions, such as reaction times (Pardo-Vazquez et al., 2019). For example, we report our success in training a model to perform a cue combination (multisensory integration; Beierholm et al. (2007)) task, where the network is trained to generate samples from the posterior of a Bayesian ideal observer. This task currently acts as a demonstration of the flexibility of our technique, rather than an extension to the mechanistic hypotheses it generates. Future work can extend in this direction.

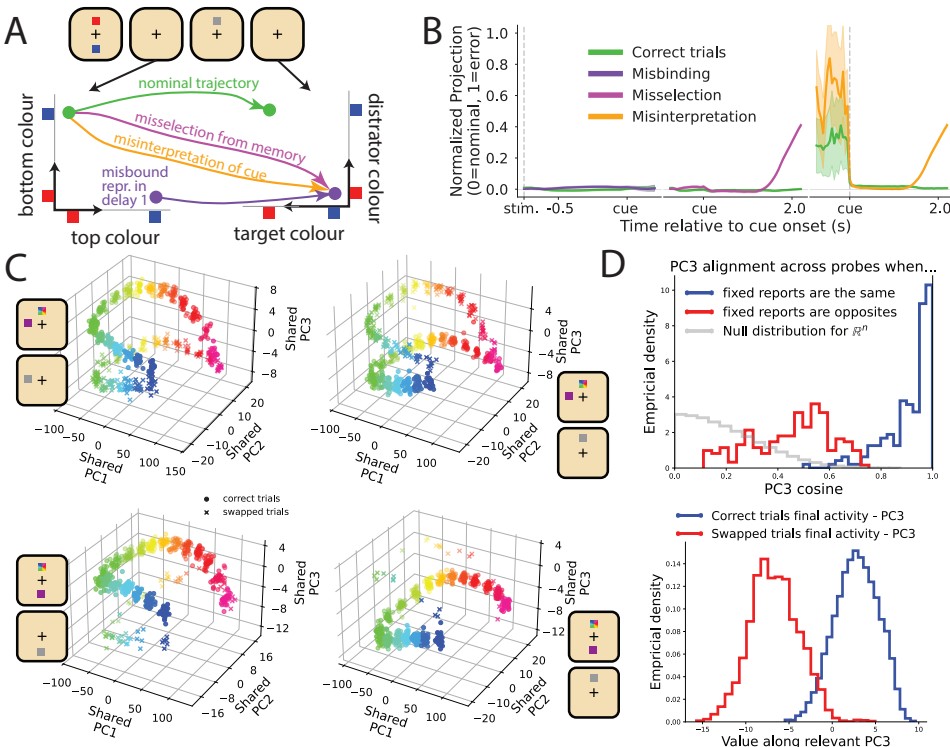

Figure 5: RNNs trained with swap errors provide a prediction about the geometry of item misselection. **A** prior works (Alleman et al., 2024) suggest 3 causes of swap errors. Only misbinding errors arise during (or before) delay 1, prior to cueing. Misselection and misinterpretation cause incorrect transfer from a probe-report binding in delay 1 to a role-report binding in delay 2. **B** We repeat linear decoder analyses from Alleman et al. (2024) for our index-cued networks. Like the real data, these provide most evidence for swap errors arising during delay 2, ruling out misbinding errors. We defer full details to Alleman et al. (2024) **C** Trial-by-trial $m_T$, projected to a shared set of principal components (PCs). One item's colour is fixed to purple, and the other item's colour is varied and used to colour the scatterplots. Left (right) axes show when the purple item is cued (a distractor), so colours indicate the uncued (target) colour. Probe features are fixed to a close (far) spatial configuration in the top (bottom) row, causing more (fewer) swap errors. Neural activities form two rings (with a gap in each due to the minimum margin between colours) - see main text. To illustrate this better, we used a network that swaps more often than seen for macaques (highest orange line in Figure 3A). **D** Left: there is high pairwise alignment in the third PC across different stimulus spatial configurations (i.e. fixed locations) if the fixed colour used to generate the two ring geometry is the same between these configurations - less so if the fixed colour is different (here, the opposite) for all pairs of spatial configurations. See Appendix E for a more graded pairwise comparison. This also applies to the first two PCs, unshown. Right: this third PC discriminates swapped versus correct trials in cases where the fixed colour item (purple here) is cued, across many different probe values.

Within swap error analysis, our analysis of the representational geometry of stimuli and responses (Figures 4 and 5) mostly considers stationary snapshots of the population activity. Besides this, Figures 4D, E aggregate information across many trials. Future work should include trial-by-trial dynamical analysis. For example, the misselection process described in Figure 5A may be due to a failure in information loading (Stroud et al., 2023) during the cue period, which is made more likely due to the higher representational alignment we uncovered here. Better interpreting why RNNs capture neural phenomena only when trained explicitly to exhibit errors, beyond the descriptive summary we have provided here, can help refine this inquiry. Looking forward, there exists a much richer set of behavioural phenomena associated with VWM which can be modelled with our method. Further dependence of swap errors on set size (Bays et al., 2009; Emrich and Ferber, 2012) and presentation order (van Ede et al., 2021; Gorgoraptis et al., 2011), and other, unimodal, response distributions which cannot be captured by moment-matching (Fritsche et al., 2020) are all suitable next paradigms to benefit from our method.

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

## A  BAYESIAN NON-PARAMETRIC MODEL OF SWAP ERRORS (BNS)

Stimulus array $Z = \{\boldsymbol{z}^{(n)} : n = 1, 2\}$ is denoted by the vector of $probe$ (location) and $report$ (colour) feature values: $\boldsymbol{z}^{(n)} = [z_p^{(n)}, z_r^{(n)}] \in S^1 \times S^1$. Item $n^*$ is cued after the first delay. The probe-dependent BNS Radmard et al. (2025) generative process defines a mixture distribution over estimates (recalled colour, $y$), based on the circular distance of the distractor from the cued item (i.e. in the probe dimension). The so-called swap function $f$ dictates the form of this dependence. For two items it is:

$$
\boldsymbol{x}^{(n)} = z_p^{(n)} \ominus z_p^{(n^*)}, n = 1, 2
$$
$$
f \sim p_f(f)
$$
$$
\tilde{\pi}^{(n)} = \begin{cases} \tilde{\pi}_{\text{correct}} & n = n^* \\ f(z_p^{(n)} \ominus z_p^{(n^*)}) & n \neq n^* \end{cases}
$$
$$
\boldsymbol{\pi} = \text{Softmax}(\tilde{\boldsymbol{\pi}})
$$
$$
\beta \sim \text{Cat}(\boldsymbol{\pi})
$$
$$
y = \varepsilon \oplus z_r^{(\beta)} \qquad \varepsilon \sim p_\varepsilon(\cdot; \phi)
$$

The associated graphical model is shown in Figure A.

To generate synthetic data, we fix one target swap function $f^*$, and perform the generative process up to component $\beta$, before embedding the samples into $\mathbb{R}^2$ and adding noise there, rather than using a circular emission distribution $p_\epsilon$. We also generate both feature values with a minimum margin of $\pi/4$, akin to previous multiitem tasks Schneegans and Bays (2017); McMaster et al. (2022):

1. Generate $Z$ with minimum feature value margins

2. Generate $y$ with BNS, using target function $f^*$

3. Embed sample and add noise $\boldsymbol{x}_T^* \sim \mathcal{N}\left( AW_x^\intercal \begin{bmatrix} \cos(y) \\ \sin(y) \end{bmatrix}, \sigma_x^2 I \right)$

The samples of $\boldsymbol{x}_T^*$ forms the target distribution for the DDPM training procedure (Appendix D).

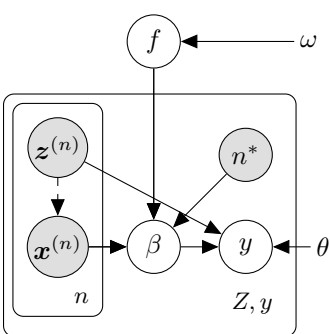

Figure 6: Graphical model of BNS, *adapted from Radmard et al. (2025).* $\theta$ contains parameters for emission distribution $p_\varepsilon$. $\omega$ contains GP prior parameters and $\tilde{\pi}_{\text{correct}}$

To fit BNS to the outputs of the trained RNN, we interpret the colour estimate as the argument of the output at the final trial timestep:

$$
y = \arctan\left( \frac{[W_x \boldsymbol{r}_T]_1}{[W_x \boldsymbol{r}_T]_2} \right) \tag{4}
$$

and use variational inference to fit $q(f)$, which approximates the dataset posterior over swap function $f$, which is equipped with a Gaussian process Rasmussen and Williams (2005) prior $p_f$. We defer details to Radmard et al. (2025). We do not consider the uniform component used in the introductory work, given the rarity of RNN behavioural samples away from the target modes.

In both cases, for the single distractor case, BNS admits a tractable probability of swapping given a probe distance:

$$p(\beta \neq n^* | \Delta z_p) = \left\langle p(\beta \neq n^* | f, \Delta z_p) \right\rangle_{p(f)} = \left\langle \frac{e^{\Delta z_p}}{e^{\Delta z_p} + e^{\tilde{\pi}_{\text{correct}}}} \right\rangle_{p(f)} \tag{5}$$

For the target swap function, this can be evaluated directly ($p(f) = \delta(f^* - f)$), forming the dotted lines in Figure 3A. For the fitted behaviour, this can be estimated using i.i.d. samples from the inferred approximate posterior ($p(f) = q(f)$).

## B   ARCHITECTURE AND TASK DETAILS

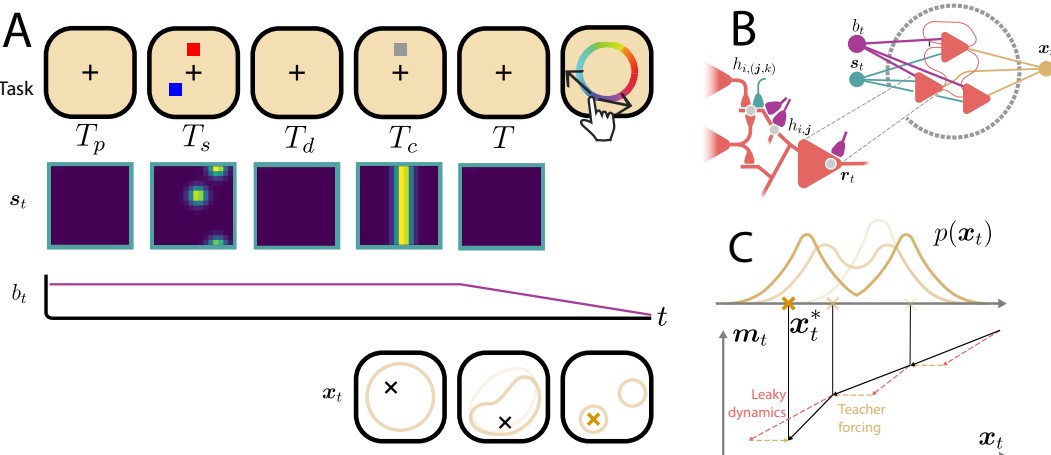

Figure 7: Schematic of architecture and training. **A** (top) task with durations; (middle) $n_s$ by $n_s$ sized conjunctive tuning curve representations of sensory inputs, and representation of time along trial (constant before denoising starts); (bottom) denoising to a mixture of Gaussians target distribution. **B** dendritic tree with sensory and time inputs. Here, $s$ in matrix form has report features (colours) for each row, and probe features (locations) for each column, hence why at cueing time, a strip column is provided, with no colour information. **C** teacher-forcing, applied to the behavioural subspace throughout the second delay period during training.

Our RNN models a population of $n$ densely connected cortical pyramidal neurons Lyo and Savin (2024). Each neuron integrates inputs from all other neurons in the population via a dendritic tree. Distal tree nodes are provided with sensory information, and all nodes are provided with a representation of time along the trial, necessary for training it as a diffusion model.

Dendritic tree nodes are organised into layers, with each node integrating activity from nodes one layer more distal from the soma than itself. Each presynaptic neuron's axons synapse onto all leaf nodes of this tree, i.e. the layer most distal to the postsynaptic neuron, including its own. This comprises a weighted sum followed by a ReLU non-linearity (Figure 7 B):

$$h_{\boldsymbol{i}}^k = \left[ \sum_j w_{(\boldsymbol{i},j)}^k h_{(\boldsymbol{i},j)}^k + b_{\boldsymbol{i}}^k(t) \right]_+ \tag{6}$$

where $k$ indexes over neurons, $\boldsymbol{i}$ indexes a particular path of nodes in the dendritic tree, and $j$ enumerates distal nodes connecting to $\boldsymbol{i}$. The population's $n$ neurons[2] receive inputs from a dendritic tree with $L$ layers, each with a branching factor of $B_l$ from the previous layer. This results in each neuron axon synapsing onto $n \prod_{l=1}^{L} B_l$ dendritic leaf nodes. The rate vector $\boldsymbol{r}$ is projected via one set of weights $W_r \in \mathbb{R}^{nB_1 \dots B_L \times n}$, then used as inputs to the most distal layer, instead of $h$. As

---

[2]Undeclared numerical constants are defined and provided values in Appendix G

mentioned in the main text, this architecture choice was partly motivated by minimum requirements of expressivity. We started with standard architectures employing linear projections followed by point non-linearities (e.g., Yang et al. (2019)), but we consistently struggled to achieve multimodal dynamics. A two-layer network at $\mathcal{F}$ could produce multimodality, but this is biologically implausible as it implies an intermediary neural population with instantaneous responses (zero time constant). We therefore adopted dendritic tree architectures (Lyo and Savin, 2024), which constrain a deep network's weights to block-diagonal form, producing tree-like connectivity that reflects cortical pyramidal neuron morphology whilst enabling the required distributional flexibility. While a dendritic structure is not necessary for our method, it is a proven (Lyo and Savin, 2024), expressive structure that maintains plausibility in this regard.

Each node is also provided with a bias term. Before the cue offset ($t < 0$), when the network is still accumulating information, this is a constant bias $b$. After cue offset, when the network is denoising during the WM second delay, this becomes a smoothly changing $n_t$-dimensional representation of time Ho et al. (2020); Vaswani et al. (2017) projected to a scalar with a unique set of weights for each dendritic node.

To provide a sensory input to the 'index-cued' network, we separate channels for stimuli and cue:

$$\boldsymbol{s}_t = \begin{cases} [z_r^{(1)}, z_r^{(2)}, \mathbf{0}^\mathsf{T}]^\mathsf{T} & t \text{ in stim. presentation} \\ [0, 0, \tilde{\boldsymbol{c}}_1^\mathsf{T}]^\mathsf{T} & t \text{ in cue presentation}, n^* = 1 \\ [0, 0, \tilde{\boldsymbol{c}}_2^\mathsf{T}]^\mathsf{T} & t \text{ in cue presentation}, n^* = 2 \end{cases} \tag{7}$$

where $\tilde{\boldsymbol{c}}_n \in \mathbb{R}^{n_c}$ are learned embeddings of each cue case.

To provide a sensory input to the 'feature-cued' network, we represent these stimuli with activations of a palimpsest conjunctively tuned neural population (Figure 7A [second row]; Matthey et al. (2015); Schneegans and Bays (2017)), then combine the tuning curve information before projection to the neural population using two sets of weights $W_p, W_r$:

$$S_t \in \mathbb{R}^{n_s \times n_s} \quad S_{t,ij} = \begin{cases} \sum_{n=1}^{2} \exp\left(\cos(z_p^{(n)} - \bar{z}_p^{(i)}) + \cos(z_r^{(n)} - \bar{z}_r^{(j)})\right) & t \text{ in stim. presentation} \\ \exp\left(\cos(z_p^{(n^*)} - \bar{z}_p^{(i)})\right) & t \text{ in cue presentation} \\ 0 & \text{otherwise} \end{cases} \tag{8}$$

$$\boldsymbol{s}_t = \text{vec}(W_p S_t W_r^t) \in \mathbb{R}^{n_i^2} \tag{9}$$

where $\bar{z}_p, \bar{z}_r$ are fixed and evenly spaced preferred stimulus values for the probe and feature dimensions. In both cases, a projection $W_s \boldsymbol{s}_t \in \mathbb{R}^{nB_1 \dots B_L}$ is provided to the most distal dendritic nodes.

The final recurrent output of the network is the somatic activity, with added noise:

$$\mathcal{F}(\boldsymbol{r_t}, \boldsymbol{s}_t, \boldsymbol{\eta}; \theta_r) = \boldsymbol{h}_\emptyset + \sigma_t^2 \eta \qquad \boldsymbol{\eta} \sim \mathcal{N}(0, I_n) \tag{10}$$

with $\theta_r$ containing all the dendritic tree weights, sensory input parameters, time embedding projection weights, etc.. $\boldsymbol{h}_\emptyset = [h_\emptyset^1, ..., h_\emptyset^n]^\mathsf{T}$ is. the somatic activity. As with time representation $b$, $\sigma_t^2$ is held at a constant maximum value before cue offset ($\sigma_{<0}^2 = \sigma_t^2$), then quenched according to the diffusion noise schedule's posterior variance (Appendix G). The overall time-discretised network dynamics are therefore:

$$\boldsymbol{r}_{t+1} = \left[\left(1 - \frac{\mathrm{d}t}{\lambda}\right) \boldsymbol{r}_t + \frac{\mathrm{d}t}{\lambda} \mathcal{F}(\boldsymbol{r}_t, \boldsymbol{s}_t; \theta_r)\right] + \sigma_t \boldsymbol{\epsilon} \quad \boldsymbol{\epsilon} \sim \mathcal{N}(0, I) \tag{11}$$

Note that additive noise is applied to the rates, rather than to the membrane activity, meaning that rate values can fall below zero. Therefore, neural rates vector $\boldsymbol{r}$ should be interpreted as deviations of the population firing rates from some baseline firing rate.

## C  SUPPLEMENTARY RESULTS FOR FEATURE-CUED NETWORKS

As mentioned in §4, while index-cued networks enjoyed a plane alignment trajectory that resembled the data with just the introduction of swap errors, probe-cued networks required higher swap errors

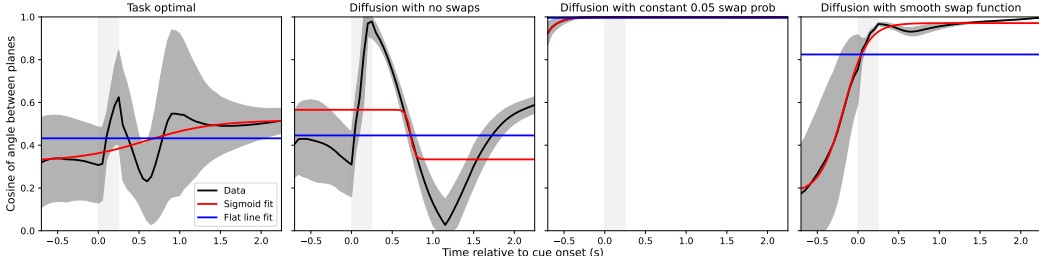

Figure 8: Sigmoid and flat line fits to the alignment trajectories of different probe-cued networks.

for proximal distractors to do so. Here, we quantify this improvement in matching trends found in the data. We fitted two models to the cosine similarity trajectories data from each network variant: (1) a four-parameter sigmoid function (slope, x-offset, y-offset, y-scale) and (2) a flat line representing the temporal mean (Figure 8). We fit these to all cosine similarity data in Figure 4D, aggregating across location distance (colours of lines). This approach was motivated by the experimental methodology in Panichello and Buschman (2021), where the authors fitted sigmoid functions to the cosine similarity over time. We adopted this as a quantitative benchmark for evaluating whether our networks produce neural dynamics that match the temporal profile observed in biological data. The comparison to a flat line serves as a control. While the constant swap rate network produces temporally stable alignments that are fit well by a sigmoid with near-zero slope, this does not match the characteristic temporal dynamics reported in the experimental results. By quantifying the mean squared error difference between sigmoid and flat line fits, we can assess which networks exhibit the dynamic temporal evolution of representational alignment seen in the biological data. These results are reported in Table 1, in the order of subplots in Figure 4D.

| | | | |
|---|---|---|---|
| Task optimal | 0.009734 | 0.005479 | 0.004255 |
| Diffusion with no swaps | 0.052094 | 0.039117 | 0.012977 |
| Diffusion with constant 0.05 swap prob | 0.000152 | 0.000000 | 0.000152 |
| Diffusion with smooth swap function | 0.065647 | 0.000340 | 0.065307 |

Table 1: Quantitative comparison of plane alignment trajectories for different feature-cued networks

## D  TRAINING DETAILS

The network trained to produce swap errors are optimised on a joint loss function. The primary terms of this loss function are the stepwise denoising errors (see below; Ho et al. (2020)), with the transition kernel mean provided by discretising the continuous RNN dynamics (equation 3) projected to the behavioural subspace. Unlike traditional DDPMs, the transition kernel takes in an $n$ dimensional rate vector, and outputs a 2 dimensional denoised mean in the behavioural subspace. Without projection $W_x$, this dictates the full network dynamics throughout the diffusion period (equation 11), but is followed by a teacher forcing step in the diffusion period during training (algorithm 2). As per the advice of seminal work on DDPMs, we weight each term of this loss equally, even though this violates its original formulation as an ELBO Ho et al. (2020).

The loss function also penalises the deviation of the distribution of the behavioural subspace activity at cue offset $x_0$ from the base (unit normal) distribution across many trials. This is a necessary starting point for the denoising process Ho et al. (2020) - because the trajectory is not teacher-forced prior to cue offset, we must explicitly regularise it. We achieve this by applying an L2/Frobenius penalty to the deviation of the first two moments of activity at the start of denoising (i.e. the first timestep of delay 2) from this target distribution Soo and Lengyel (2022). Finally, we apply an L2 regulariser on neural activity across the trial, including prior to the denoising period Yang et al. (2019); Stroud et al. (2023).

Normally, training DDPM parameters across different timesteps, i.e. $p_\phi(\cdot|\cdot',\tau)$ can be done in parallel across $\tau$: data samples $x_T$ are noised to a valid sample of $q(x_\tau|x_T)$ with the noising process

accumulated over timesteps, then the DDPM objective can be calculated independently of the noised state at other timesteps. This is because DDPMs are Markovian - trained parameters $\phi$ can denoise a sample independently of previous timesteps. This is not possible when training RNNs to perform memory tasks. The necessary computations - memory maintenance during the delay period and retrieval during the cueing period - are implemented via non-linear *dynamical motifs* (Vyas et al., 2020; Driscoll et al., 2022; Versteeg et al., 2025) which are also driven by the remainder of the neural state space which is not projected to the response space. As such, training RNNs requires simulation of full task trials in sequence. As neural activity trajectories will likely veer out of the sequence of distributions defined by the noising process $q$, this sequential training risks detrimentally biasing the training distribution. To solve this, we use *teacher-forcing* to ensure the correct distribution of training targets is satisfied (Appendix D; Williams and Zipser (1989)).

---

**Algorithm 2** Training with teacher forcing. Here, $\bar{r}, \Sigma_r$ are the empirical mean and covariance of $r_0^k$ across many trials $k$ with the same experimental conditions presented.

---

**repeat**
    Generate stimulus set and cue $(z^{(1)}, z^{(2)}, n^*)$
    **for** each independent trial, indexed by $k$ **do**
        From BNS, generate a sample from the desired behavioural distribution: $x_T^{*,k}$, and apply the noising process (equation 2)[3] to generate noised data $x_{0:T-1}^{*,k}$
        Initialise network activity $r^k \sim \mathcal{N}(0, \sigma_r^2 I)$
        Run network dynamics for the prestimulus, stimulus, delay, and cue periods (Figure 7A, Appendix B) with noisy discretised leaky dynamics (equation 3 without projection), finally arriving at network activity at cue offset $r_0^k$
        **for** $t = 0$ **to** $T$ **do**
            Replace behavioural subspace: $r_t^k \leftarrow r_t^k + x_t^* - W_x W_x^{\mathsf{T}} r_t^k$
            Calculate mean transition kernel $\hat{\mu}_{\theta_r^k}(r_t, t)$ (equation 3) and stash
            Run dynamics one step (equation 11) and add scheduled process noise (Appendix G).
        **end for**
        Train on joint loss function

$$\sum_k \left[ \sum_{t=0}^{T} \|\mu_q(x_t^{*,k}, x_T^{*,k}) - \hat{\mu}_{\theta_r}(r_t^k, t)\|_2^2 + \gamma_2 \sum_{\text{all timesteps } s} \|r_s^k\|_2^2 \right] + \gamma_1 \left( \|\bar{r}_0\|_2^2 + \|\Sigma_{r_0}\|_{\text{Fr}}^2 \right)$$

    **end for**
**until** converged

---

### D.1 Changing optimisation window for task-optimal networks

Task optimal networks are trained on a simple MSE loss function:

$$\sum_{\tau=0}^{T^*} \left\| A W_x^{\mathsf{T}} \begin{bmatrix} \cos(z_r^{(n^*)}) \\ \sin(z_r^{(n^*)}) \end{bmatrix} - W_x r_{T-\tau} \right\|_2^2 \tag{12}$$

i.e. they are trained to minimise distance to the point on a circle in $\mathbb{R}^2$ with the same argument as the task-optimal colour estimate $z_r^{(n^*)}$. Training with a similar, output-based MSE loss function against a multimodal target distribution would lead to mode-averaging, warranting the process-based MSE loss we have developed in this work. This is illustrated in Figure 9, which shows samples generated from a network trained in this way.

Note that only the final network rate vector $r_T$ is behaviourally relevant, as it is used to evaluate the colour estimate of the network for that trial (Appendix A). We refer to the number of timesteps $T^*$ before this critical timestep which the optimisation kicks in as the optimisation window. We maintained the same duration and neural noise schedules of each trial period across all networks (Appendix G). For a different, simpler VWM task with a single, variable delay duration, it has previously been shown that adjusting the optimisation window alters the neural coding schemes used for VWM Stroud et al. (2023). Specifically, it affected the cross-temporal decodability of task

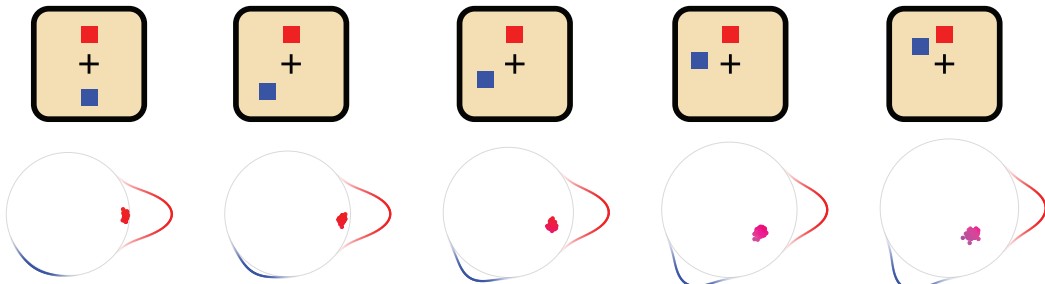

Figure 9: Network trained on a MSE loss (equation 12) against samples from a multimodal target distribution of estimates. The samples are coloured by colour estimate which they are interpreted as, and the modes of the target distribution for each trial is coloured by the nominal colour estimate at which it is centered. As the probe feature values get closer, the blue distractor is more likely to cause a swap error, as reflected in the target distributions. The target multimodality is not achieved, instead the network interpolates between the two distinct modes.

variables in storage for the duration of the delay before the estimate is produced. In the absence of neural evidence to compare to for the neural coding scheme across this delay Panichello and Buschman (2021), we presented the extreme case of $T^* = T$ Figure 4E (left), i.e. the MSE loss function in equation 12 was applied to all timesteps of the second delay. Figure 10 shows the equivalent plot for multiple task-optimal networks with different optimisation windows. In all cases, All estimates were tightly bound to the circle in the behavioural subspace, and no swap errors were observed (Figure 3C [top]). Again, we see that the hallmarks of neural data are not captured, although in networks which are afforded time before being penalised for behavioural deviation, there is a shared alignment in the target-distractor colour planes during the optimisation window. Regardless, without swap errors, it is difficult to make connections to behavioural data, as was the original aim of our novel training method. Investigating the link between this property of neural activity and optimal information loading is key to understanding both the mechanism of memory retrieval, and the dynamical source of swap errors during this time (§5), and is left to future works.

## D.2 TRAINING CURRICULUM

Training feature-cued networks directly as described above, be it on task-optimality or behavioural realism, proved inefficient and time-costly. To speed up training, feature-cued networks were first trained on a much shorter version of the task, with a second delay of $T = T^* = 5$ timesteps. This ensured that networks were initialised with the ability to, at least, retrieve cued items from memory. From this checkpoint, each network type was then trained independently to prevent cross-contamination of the specialised mechanisms we sought to develop and study in each model variant.

## D.3 COMPUTATIONAL RESOURCES

All models were trained using a single NVIDIA RTX A5000 24GB GPU at a time. All optimisation is implemented in `PyTorch`, utilising the Adam optimiser. A learning rate of 0.001 was found satisfactory across all experiments presented, and other optimiser hyperparameters were maintained at the library defaults. A batch size of 32, each with 512 independently simulated trials (indexed by $k$ in Algorithm 2), was used throughout.

Index-cued networks trained for task optimality (behavioural realism) typically took 35000 (300000) batches at 3.0 (3.0) batches per second, resulting in 3.2 (28) hours per run. These experiments typically occupied 15650 MiB of VRAM. Feature-cued networks trained for task optimality (behavioural realism) typically took 2000[4] (300000) batches at 2.4 (2.3) batches per second, resulting in 0.23 (36)

---

[4]Compared to the training preamble (see later in main text) this was just a matter of extending time horizons, rather than learning a new memory retrieval mechanism from scratch

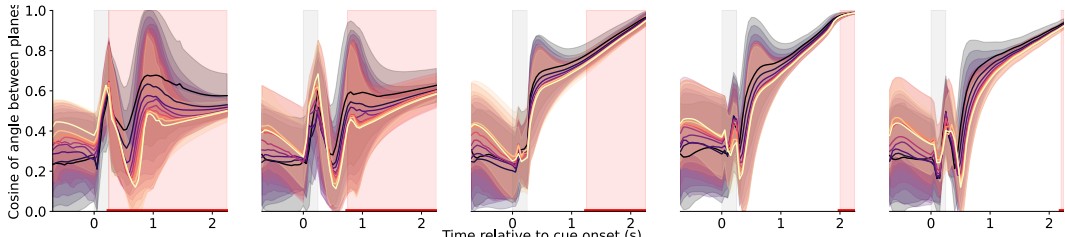

Figure 10: Plane alignment for different task-optimal networks (see appendix text). Optimisation window is highlighted in red

hours per run. These experiments typically occupied 15870 MiB of VRAM. The training preamble with a shortened trial duration was typically took 500000 batches at 8.6 batches per second, resulting in an additional 16 hours per initialisation; but this was inherited by multiple downstream networks with different final objectives. These experiments typically occupied 4560 MiB of VRAM. The reduction in memory is due to the shorter time per trial in this preamble.

## E    NEURAL SIGNATURES OF SWAP ERRORS

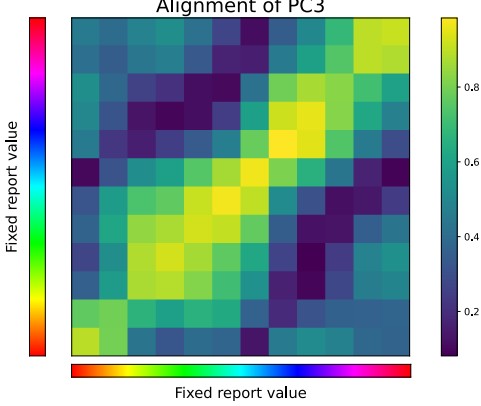

Figure 11: Continuing from Figure 5C – cosine between PC3 for response time activity geometries generated by different fixed colours, averaged over spatial configurations (spatial pairs).

Figure 11 provides an extension to the alignment results of Figure 5C (left). There, the item locations are kept fixed, as well as one item's colour. The response-time hidden activity $m_T$, when projected to its first three principal components (PCs), forms a characteristic 2 ring structure. These PCs are shared across initial item locations, as long as the fixed colour is the same (blue histogram). This is not the case when the colours are dissimilar (red histogram). Figure 11 provides the mean value of this histogram for a wider range of fixed colour pairs. We see a cyclical structure of PC3 alignment with respect to the fixed stimulus colour. This provides preliminary evidence of a toroidal structure, which would have to be verified with further topological data analysis Cueva et al. (2021).

## F    EXTENSION TO CUE COMBINATION TASKS

### F.1    TASK AND GENERATIVE MODEL

To demonstrate the generality of our approach beyond visual working memory, we trained networks on a causal inference task for multisensory cue combination. In this task, an observer receives visual

$(s_v)$ and auditory $(s_a)$ cues, and must estimate the location of the auditory source $z_a$ (Figure F.3.2a). The cues may or may not originate from a common source. When they do share a common cause, integrating both cues improves estimation accuracy, but when they do not, the visual cue acts as a distractor that should be ignored.

We generated synthetic behavioural data using the Bayesian causal inference model from Beierholm et al. (2007), which has been successfully fit to human psychophysical data in this task. The generative model assumes that visual and auditory sources ($z_v$ and $z_a$) are drawn independently from a Gaussian prior with standard deviation $\sigma_p$. On each trial, these sources either share a common cause ($C = 1$) with probability $p_{\text{common}}$, or are independent ($C = 0$). When $C = 1$, both cues are generated from the same underlying location; when $C = 0$, they are generated independently. The cues are corrupted by Gaussian sensory noise with standard deviations $\sigma_v$ and $\sigma_a$ respectively (Figure F.3.2c). The ideal observer computes the posterior distribution over $s_a$ by marginalising over the latent causal structure:

$$p(s_a|x_v, x_a) = p(C = 1|x_v, x_a)p(s_a|x_v, x_a, C = 1) + p(C = 0|x_v, x_a)p(s_a|x_a, C = 0) \quad (13)$$

Following the original work, we used parameter values $\sigma_v = 2.14$, $\sigma_a = 9.2$, $\sigma_p = 12.3$, and $p_{\text{common}} = 0.28$. Spatial standard deviations were scaled by a factor of $1/12.3$ to match the spatial scales used in our network training. The resulting posterior distributions are typically bimodal when the cues are separated (see Figure F.3.2a for examples with fixed $x_a = 0$), with one mode corresponding to integration (visual cue has a common cause) and another to segregation (visual cue has an independent cause and should be ignored).

In summary, the generative model assumes:

$$z_a \sim \mathcal{N}(0, \sigma_p^2) \quad (14)$$

$$C = \begin{cases} 0 & \text{w.p.} \quad p_{\text{common}} \\ 1 & \text{w.p.} \quad 1 - p_{\text{common}} \end{cases} \quad (15)$$

$$z_v \begin{cases} = z_a & \text{if } C = 1 \\ \sim \mathcal{N}(0, \sigma_p^2) & \text{if } C = 0 \end{cases} \quad (16)$$

$$s_a \sim \mathcal{N}(z_a, \sigma_a^2) \quad (17)$$

$$s_v \sim \mathcal{N}(z_v, \sigma_v^2) \quad (18)$$

## F.2 NETWORK TRAINING

Unlike the circular feature spaces used in the VWM tasks (§3), the cue combination task involves linear spatial locations. We therefore adapted our architecture to receive 2D sensory inputs $[s_v, s_a]$ during the stimulus presentation epoch, and to output 1D estimates of auditory location $z_a$ via a single output dimension (Figure F.3.2b). The trial structure comprised a prestimulus baseline period, followed by simultaneous presentation of both cues, then a delay period during which the network performs diffusion-based denoising to generate an estimate.

We used the same dendritic tree RNN architecture as in the feature-cued VWM networks, with identical regularisation and training hyperparameters. The key difference was in the dimensionality of the behavioural subspace – here, a 1D projection $x = w_x^\top r$ rather than the 2D projection used for circular colour reports. As in the VWM case, the network was trained using teacher-forcing during the delay period to denoise from the corruption process to synthetic behavioural samples. In this case, these samples were drawn from the trial-specific posterior distribution $p(z_a|s_v, s_a)$, as infered by an ideal observer (i.e., with access to the real generative model parameters). This posterior was discretised into 200 bins spanning $[-4\sigma_p, +4\sigma_p]$, and samples were drawn by first selecting a bin according to the posterior probability mass, then adding uniform noise within that bin to generate continuous-valued targets.

## F.3 RESULTS

### F.3.1 MODEL CAPTURES IDEAL OBSERVER POSTERIORS

We verified that our trained network successfully reproduces the ideal Bayesian observer posteriors across different cue configurations. Figure F.3.2 shows results for the case where the auditory cue

is fixed at its prior mean ($s_a = 0$) while the visual cue location $s_v$ varies from $-3$ to $+3$. The ideal observer posterior (Figure F.3.2a) exhibits characteristic structure: a strong horizontal streak at $z_a = 0$ (corresponding to the auditory cue location) and a diagonal streak interpolating between $z_a = 0$ and $z_a = x_v$ (corresponding to the visual cue location when integrated under the common cause hypothesis). The relative strength of these two modes is determined by the posterior probability of a common cause, $p(C = 1|x_v, x_a)$, which decreases as the cues become more separated (Figure F.3.2c).

When $x_v$ and $x_a$ are similar (left side of heatmap), the posterior is dominated by the common cause hypothesis, producing a single integrated estimate that combines both cues with reliability-based weighting. As the cues diverge (right side of heatmap), the posterior becomes increasingly bimodal, with the segregation mode (at $s_a = x_a = 0$) gaining probability mass while the integration mode (along the diagonal) weakens. This bimodality reflects the observer's uncertainty about whether the cues share a common cause.

The network successfully reproduces this structure (Figure F.3.2b), as evidenced by histograms of samples generated by the network. Example slices (Figure F.3.2c) show close correspondence between the target posteriors (black lines) and empirical sample distributions (colored histograms) across a range of visual cue locations, capturing both the unimodal regime when cues are close and the bimodal regime when they are separated.

### F.3.2 QUANTIFYING ACCURACY ACROSS THE FULL CUE SPACE

To comprehensively evaluate the network's performance, we computed the Kolmogorov-Smirnov (KS) statistic comparing the ideal and empirical posterior distributions across a $20 \times 20$ grid spanning all combinations of $x_v \in [-3, 3]$ and $x_a \in [-3, 3]$ (Figure F.3.2a). The KS statistic quantifies the maximum vertical distance between two cumulative distribution functions (CDFs), with values near zero indicating close agreement.

The heatmap reveals that the network achieves high accuracy (low KS statistics, shown in blue) across most of the cue space. Example CDF comparisons (Figure F.3.2b) illustrate this agreement for representative conditions spanning the range of KS values. For each comparison, the ideal CDF (black line) closely tracks the empirical CDF computed from network samples (coloured line), with the maximum discrepancy (red vertical line) typically small. Cases with slightly higher KS statistics tend to occur when cues are moderately separated – a regime where the posterior is most strongly bimodal and thus most challenging to sample accurately.

Figure 12: Causal inference for multisensory cue combination. **A** Task schematic: the observer receives visual ($s_v$) and auditory ($s_a$) cues and must estimate the auditory source location $z_a$. Cues may share a common cause. **B** Trial timeline for the RNN. After a prestimulus phase, both cues are presented simultaneously, followed by a delay period during which the network denoises to produce an estimate. Unlike the VWM tasks, sensory inputs are 2D $[s_v, s_a]$ and output is 1D (estimate of auditory location - fit to be a sample of the ideal observer posterior). **C** Bayesian graphical model. Sources $z_v$ and $z_a$ are drawn from a common prior, and the causal structure $C$ determines whether they are bound together or independent. Cues $s_v$ and $s_a$ are noisy observations of the sources.

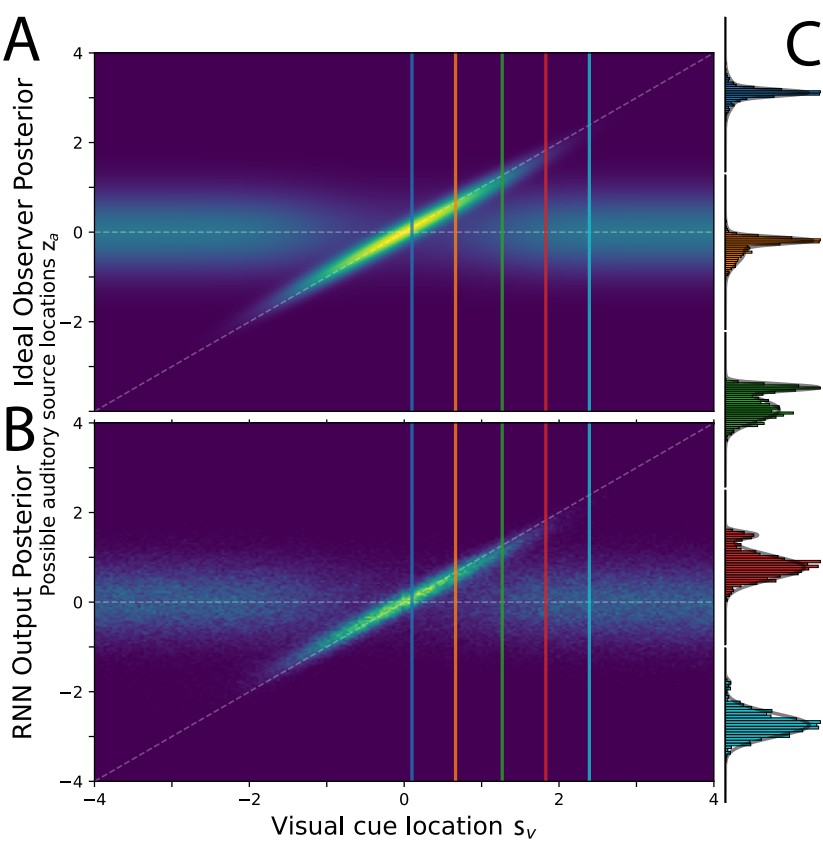

Figure 13: Network reproduces ideal observer posteriors for fixed auditory cue. **A** Ideal observer posterior $p(z_a|s_v, s_a)$ across varying visual cue locations $s_v$ with fixed auditory cue at $s_a = 0$ (dashed white line). The diagonal streak (dashed white line) corresponds to the shared source hypothesis, while the horizontal streak corresponds to the separate sources hypothesis. **B** Empirical posterior computed from network samples shows close correspondence to ideal observer. **C** Example slices at selected visual cue locations (colored vertical lines in panels a,b) showing target posterior (black line) and sample histograms (colored). Network successfully captures both unimodal and bimodal regimes.

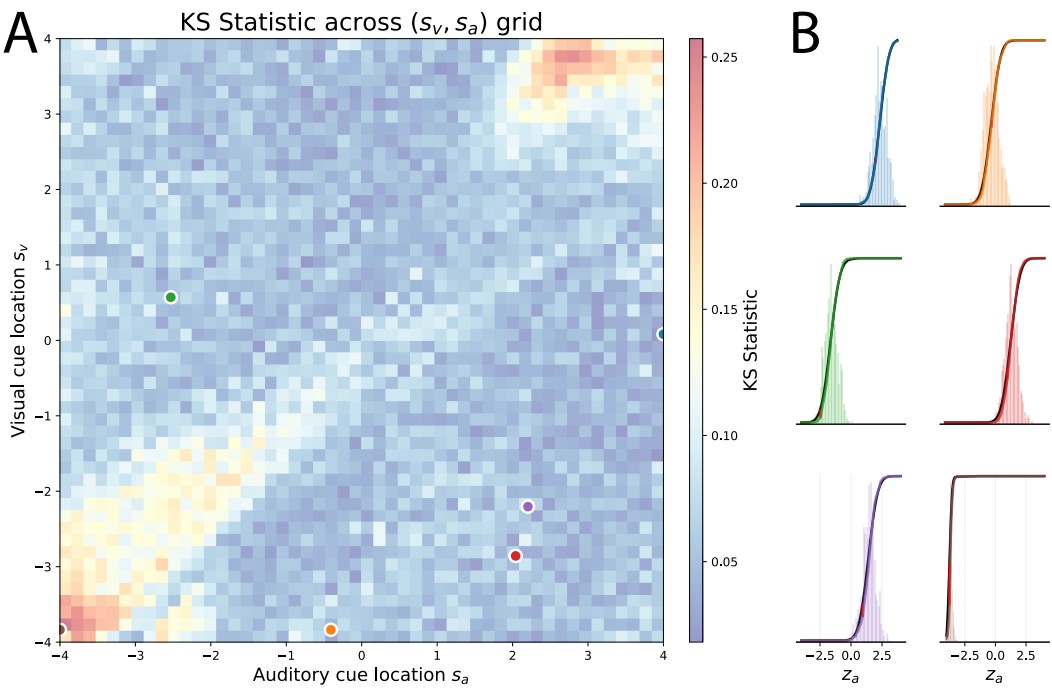

Figure 14: Kolmogorov-Smirnov analysis across full cue space. **A** KS statistic quantifying agreement between ideal and empirical posteriors across all combinations of visual ($x_v$) and auditory ($x_a$) cue locations. Low values (blue) indicate close agreement. Colored dots mark locations of example comparisons shown in panel b. **B** Example CDF comparisons for representative conditions spanning the range of KS values. Each panel shows the target posterior CDF (black), the empirical sample CDF from network samples (colored), and the maximum discrepancy (red vertical line). Network achieves high accuracy across diverse cue configurations.

## G  TABLE OF CONSTANTS AND NOISE SCHEDULE

| Name | Symbol | Value |
|---|---|---|
| Minimum angular margin between item feature values | - | $\pi/4$ |
| Radial magnitude of behavioural target | $A$ | 2.5 |
| Variance of behavioural target models | $\sigma_x^2$ | $0.2^2$ |
| Variance of rate vector initialisation target models | $\sigma_r^2$ | $1.0^2$ |
| Network population size | $n$ | 16 |
| Number of dendritic tree layers | $L$ | 2 |
| Branching factor of each dendritic tree layer | $B_1, B_2$ | 10, 10 |
| Dimensionality of global time representation | $n_t$ | 16 |
| Shared time representation vector size | - | 8 |
| Number of tuning curves for each feature dimension | $n_s$ | 16 |
| Feature-cued sensory projection size | $n_i^2$ | $6^2$ |
| Index-cued cue embedding size | $n_c$ | 4 |
| Time discretisation step | $dt$ | 0.05 |
| Neural membrane time constant | $\lambda$ | 0.5 |
| Prestimulus duration | $T_p$ | 0.15s, 3 timesteps |
| Stimulus exposure duration | $T_s$ | 0.25s, 5 timesteps |
| Delay 1 duration | $T_d$ | 0.75s, 15 timesteps |
| Cue exporsure | $T_c$ | 0.25s, 5 timesteps |
| Delay 2 (denoising) duration | $T_d$ | 2.0s, 40 timesteps |
| Base distribution regularisation weight | $\gamma_1$ | 0.01 |
| L2 rate regularisation weight | $\gamma_2$ | 0.0001 |

Table 2: Constants and hyperparameters used in main text

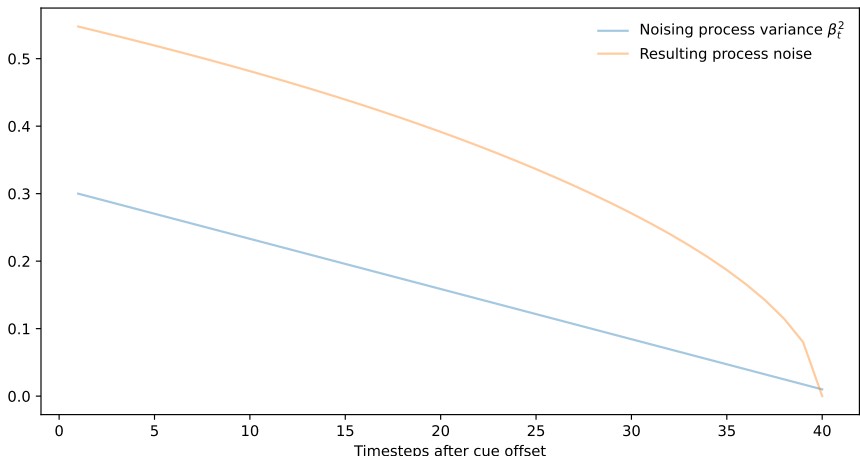

Figure 15: Schedule for corruption noise variance ($\beta_t$; equation equation 15) and corresponding generative noise standard deviation ($\sigma_t$). The former was a linear interpolation between 0.3 and 0.01 across the 40 timesteps.

