# OpenReview forum: "Setting up for failure: automatic discovery of the neural mechanisms of cognitive errors"
_ICLR.cc/2026/Conference — ICLR 2026 Poster_

### Official Review · Reviewer_YEvN · 2025-10-15

**Soundness:** 3
**Presentation:** 2
**Contribution:** 3
**Rating:** 4
**Confidence:** 3

**Summary:**

The paper proposes training RNNs to reproduce behavioral distributions including characteristic error modes than optimizing task performance. They do so by first using a generative model to produce synthetic data for training RNNs, then adapting diffusion training procedures for RNNs to produce realistic behavioral distributions. The authors tested their method on the visual working memory task, and found that the DDPM-trained RNNs, instead of the task-optimized RNNs, produces realistic swap errors and neural representations.

**Strengths:**

1. Training RNNs to produce realistic and complex behavioral distributions is a very important and timely research direction.
2. This paper is very well-written and easy to follow.
3. The experiments are well designed and the authors provided analysis at both the behavioral and representational level.

**Weaknesses:**

1. The BNS model used to generate synthetic data still requires hand-tuning, so “automatic discovery” feels overstated. Could the authors comment on this limitation?
2. In Figure 3A, the correspondence between border colors and RNN models is unclear. Please state this explicitly in the caption (or add a legend).
3. On line 203, (F) has not been introduced, and the network equation is missing. It would help to present Equation 3 (or at least the network update equation within it) earlier.
4. On line 252, I think the intended expression is (x = G(r) = W_x r) (the r appears to be missing).
5. The approach depends on a generative model to create synthetic behavioral data for training the RNNs, which raises several concerns:

   * For more complex tasks, well-founded hypotheses or statistical/generative models of the behavioral process may be unknown or unavailable.
   * The method assumes the generative model captures key characteristics of behavior, yet the paper provides no quantitative evidence for this beyond showing that swap error increases when the probe is closer to the cued item.
   * Lines 146–149 claim that models listed in §1 “can be used to faithfully generate synthetic behavioural data,” but this is not supported with empirical evidence or citations to prior work.

If these concerns are properly addressed, I may consider increasing my score.

**Questions:**

1. Does well does the method generalize to a more complex versions of the task, for example, with more items?
2. How does the method apply to tasks where an output is required at every time step of the trial?
3. Fig 3C: for the brown-border model, shouldn’t close probe feature values have more swap error? (It looks like it's the opposite in the figure).

---

> ### Author Response · Authors · 2025-11-23
>
> We sincerely thank the reviewer for their positive assessment of our work and for highlighting the importance and timeliness of our research direction. We particularly appreciate the reviewer's openness to reconsidering their score if the raised concerns are properly addressed, and we have worked to provide the clarifications and additions requested (highlighted in blue).
>
> **On generalisation to more complex tasks:**
>
> We have added substantial new material demonstrating the flexibility of our approach:
>
> 1. **New task domain** (Appendix F): We present results from a multisensory cue combination task, demonstrating that our method successfully captures complex posteriors where mode statistics have more complex dependencies on cue locations. Quantitative validation via CDF comparison (Appendix F.3.2) confirms high accuracy across the full stimulus space.
> 2. **Within-domain extensions**: Whilst we successfully trained networks on 3+ item VWM tasks to reproduce the target behavioural distribution, we were unable to identify consistent neural geometries without corresponding neural data for validation. We leave this direction of generalisation to future research, and instead focussed on generality across task domains (VWM and multisensory integration) for this work.
> 3. **Limitations**: We acknowledge that our current implementation does not immediately extend to tasks requiring continuous behavioural outputs at every timestep (e.g., motor control). We look to diffusion-based methods from the RL literature (which sample from full trajectory distributions) to enable such extensions in future work.
>
> **On the dependence on generative models:**
>
> Whilst we agree that our method requires a generative behavioural model, we note that:
>
> 1. For many well-studied cognitive tasks, such models already exist and have been successfully fit to data (see introduction and discussion), but lack a dynamical, mechanistic counterpart. Our method allows computational neuroscientists to fill this gap.
> 2. We selected BNS because it flexibly captures swap error dependencies that previous normative and non-dynamic neural models missed (Radmard et al., 2025). However, we must clarify BNS's role in our work: we only used BNS's fitting capability on data *generated* by our trained networks, not on real behavioural data. The real behavioural data we matched was a simple \~10% swap rate in a binary-location task. BNS served purely as a tool to generate data and compare our trained networks' output distributions to their target distributions (Figure 3A).
> 3. Where novel tasks lack established models, recent advances in descriptive modelling (e.g., Binz et al., 2024\) provide increasingly powerful tools for capturing complex behavioural distributions, a lower bar than providing mechanistic or normative details. We have clarified this in the revised manuscript.
>
> **On BNS hand-tuning and "automatic discovery":**
>
> Our approach follows a principled validation strategy: we first establish correspondence with available neural data (index-cued networks compared to macaque data from Panichello & Buschman, 2021), demonstrating that our method recovers known neural geometries when trained on appropriate behavioural distributions. To this end, we have also included new linear decoder analyses for the index-cued networks (Figure 5B), again replicating results from the macaque experiments.
>
> Having validated this correspondence, we then extend to feature-cued networks using archetypal swap dependencies derived from human psychophysical experiments (e.g., McMaster et al., 2022), which consistently show that proximity increases swap errors. Whilst we lack neural recordings for these feature-cued variants, the established validity of our approach with index-cued networks gives us confidence that the resulting predictions about neural geometry (Figure 4D, 5\) are meaningful and testable. We use archetypal swap dependencies in these cases, but also show that our method can recover behavioural distributions for a range of these dependencies (Figure 3).
>
> **Addressing specific weaknesses:**
>
> * **Figure 3A colours**: We have clarified their relationship in the caption.
> * **Line 203 (Equation 1\)**: The function *F* was introduced earlier in Equation 1, but we now include an explicit forward reference in Section 3 to aid reader navigation. We deliberately deferred presenting Equation 3 until after introducing our DDPM-based training approach, as the equation's interpretation depends on understanding this context.
> * **Line 252**: We have corrected the missing *r* in the equation.
> * **Figure 3C**: Thank you for catching this error, we have corrected the stimulus schematics, which were indeed, ironically, swapped.
>
> With these substantial additions, we hope the reviewer will consider increasing their score. We believe we have adequately addressed all major concerns whilst being forthright about the method's current limitations and future directions.

---

> > ### Comment · Reviewer_YEvN · 2025-11-24
> >
> > I appreciate the new task & analysis added to the revised manuscript, and the clarification provided by the authors in their rebuttal. I'm happy to increase my score to 6.

---

### Official Review · Reviewer_96uo · 2025-10-27

**Soundness:** 2
**Presentation:** 3
**Contribution:** 2
**Rating:** 4
**Confidence:** 3

**Summary:**

This work trained RNNs to exhibit swap errors during a multi-item delayed estimation VWM task. Its success was contrasted with naive training methods applied to task-optimal networks. The proposed method captured a more accurate description of neural mechanisms when trained to adhere to the empirical variability around real swap errors.

**Strengths:**

1. The research problem itself is interesting and promising, i.e. discovering neural mechanisms of cognitive errors. However, it is doubted whether such simulated errors by RNNs can be really called 'neural mechanisms' of human cognitive errors, since the mechanisms were revealed in RNNs rather than human neural activities.

2. The background section makes it easier for the reader to understand the context.

3. The proposed model was evaluated in a working memory task dataset and showed meaningful patterns and signatures of cognitive behaviors.

**Weaknesses:**

1. Most evaluations are qualitative results, rather than quantitative results. Although the discovered signatures are promising, the lack of quantitative measurements significantly reduces the rigor of this work.

2. Only one task (working memory) is used for evaluation. Although I agree that this is a representative task, the small task domin limits the generalization ability of this proposed model and it is not convincing that it can be effective in other cognitive task domains.

3. Limited baseline or ablation studies to show the unique advantages of the proposed model to discover mechanisms of cognitive errors compared with prior models in existing work. I can only find one naive training method as the baseline.

4. Lack of details in the dataset used for evaluation, including working memory task settings/design, sample size of participants and trials in the task, etc. This makes it hard to evaluate the significance of this work and reliability of the findings.

**Questions:**

Please check my concerns and questions in Weaknesses section.

---

> ### Author Response · Authors · 2025-11-23
>
> We sincerely thank the reviewer for their thoughtful evaluation, and encouraging remarks about the promise of our research problem.
>
> **On the concern about "simulated errors" and neural mechanisms:**
>
> We appreciate the opportunity to clarify our methodological approach. The structure of our results is specifically designed to address this concern. In each subsection, after establishing our method's ability to match complex target distributions (Figure 3), we compare our networks to macaque neural data where available (Figure 4B-C), demonstrating similarities in neural geometry. To this end, we have also included new linear decoder analyses for the index-cued networks (Figure 5B), again replicating results from the macaque experiments. We then extend these findings to make testable predictions (Figures 4D, 5).
>
> Our aim is to establish that by fitting to detailed behavioural data, we create neural models that match existing experimental analyses, then leverage these models to generate predictions for future experimental investigation. We view these predictions not as definitive explanations, but as mechanistic hypotheses to be validated through subsequent experimental work. This approach offers a principled, data-driven path from behaviour to neural mechanism that complements traditional experimental methods.
>
> **On qualitative results and task scope:**
>
> We have substantially extended our work to address these concerns. The revised manuscript now includes (highlighted in blue):
>
> 1. **New task domain**: We present results from a multisensory integration task for cue combination in Appendix F. While we do not make claims about neural activity for this task, we demonstrate that our method successfully reproduces complex, multimodal behavioural distributions in this entirely different cognitive domain, with quantitative validation metrics.
> 2. **Quantitative analyses**: We now provide quantitative comparisons of plane alignment trajectories across network variants. Specifically, we fit sigmoid functions to alignment dynamics (following the methodology of Panichello & Buschman, 2021\) and compare these to flat-line baselines. The MSE differences (Table 1 in Appendix C) demonstrate that only networks trained with realistic, distance-dependent swap errors capture the characteristic temporal dynamics observed in biological data. Additionally, for the multisensory task, we provide Kolmogorov-Smirnov statistics quantifying distributional accuracy across the full stimulus space (Appendix F.3.2).
>
> **On the lack of baseline comparisons:**
>
> We cannot identify any comparable dynamical, mechanistic models capable of generating multimodal response distributions such as swap errors. The existing literature either produces: models that capture swap errors but lack recurrent neural dynamics (e.g., Schneegans & Bays, 2017; McMaster et al., 2022), or dynamical RNN models that are task-optimised but fail to exhibit swap errors altogether (e.g., Yang et al., 2019; Driscoll et al., 2022). To our knowledge, no prior work has successfully trained recurrent neural networks to produce the characteristic multimodal response distributions that arise from swap errors in multi-item working memory.
>
> Figure 3D-E demonstrates that intuitive modifications to task-optimal networks (varying noise levels, manipulating stimulus separation) fail to produce swap errors. Achieving these errors through traditional approaches would require iterative, heuristic ablations—precisely the manual, piecemeal process our method aims to circumvent. We show (Figure 4, Figure 9\) that only training with realistic error distributions captures both behavioural and neural signatures, representing a fundamental advance over prior approaches. This represents the key novelty of our contribution: bridging the gap between flexible behavioural modelling and biologically plausible neural dynamics.
>
> **On dataset information:**
>
> We apologise for the insufficient detail. We now clearly state (highlighted in blue, Section 4\) that we use the publicly available dataset from Panichello & Buschman (2021), published in *Nature*. We have added relevant statistics: monkey 1 contributed 3,943 completed trials over 10 sessions; monkey 2 contributed 4,769 trials over 13 sessions, with a pooled swap error rate of approximately 10%.
>
> With these substantial additions, including a new task domain, quantitative validation metrics, clarification of our methodological uniqueness, and complete dataset documentation, we believe we have addressed all major concerns raised. We hope the reviewer will consider increasing their score in light of these improvements to the manuscript.

---

> > ### Comment · Reviewer_96uo · 2025-11-24
> >
> > I appreciate the detailed clarification and new tasks with analysis. I'm happy to increase my score. Thanks!

---

### Official Review · Reviewer_eTxk · 2025-10-31

**Soundness:** 3
**Presentation:** 2
**Contribution:** 3
**Rating:** 6
**Confidence:** 3

**Summary:**

The authors train RNNs to fit distributions of behavioral output. This is done by modifying a diffusion based procedure. The algorithm is tested on a synthetic model of behavior. The resulting network’s hidden state statistics has similarities to those observed experimentally.

**Strengths:**

Behavior is often probabilistic, and not necessarily unimodal. Most models add some noise on top of a deterministic backbone. Using flexible distributions directly is an important contribution.

**Weaknesses:**

Despite the introduction, there is no fit to actual behavioral data. Is there any guarantee that this can work?

Clarity: There are several places where the writing is not clear. For instance, the paper jumps between index and feature based codes, without explaining the rationale.

The basic premise in line 41: RNNs are not the only option for testable hypotheses. They are a specific example of a latent variable model.

Line 110 – biologically plausible is a broad and ill-defined term. Some people only consider spiking networks to be plausible. Others require conductance based models. Others are satisfied with rate models, but insist on Dale’s law.

**Questions:**

Line 120 – Why dendritic tree?
Lines 208-214 The description of index and feature based is not written clearly. The overuse of the word feature does not help.
Line 253: Clarity. If I understand correctly, the network activity r is split. But the sentence reads as if it is split into x and m. The next sentence reads “this output”, but the word output does not appear in the previous sentence.
Figure 7A: could be useful to explain what the horizontal and vertical axes of s_t are

---

> ### Author Response · Authors · 2025-11-23
>
> We sincerely thank the reviewer for their positive assessment and for recommending acceptance of our work. We are pleased that the reviewer recognises the importance of our contribution to directly fitting flexible behavioural distributions.
>
> **On clarity regarding index vs feature-based coding:**
>
> We appreciate this observation and have substantially improved the manuscript to make this progression more explicit (highlighted in blue). Our rationale, now clearly stated at the beginning of each results subsection, is as follows:
>
> We begin with index-cued networks because they directly correspond to the experimental task for which we have neural data, allowing validation of our approach by establishing that our method recovers known neural geometries. To this end, we have also included new linear decoder analyses for the index-cued networks (Figure 5B), again replicating results from the macaque experiments. We then extend to feature-cued networks to make predictions about how continuous probe features (locations) affect neural representations.
>
> **On fitting to actual behavioural data:**
>
> We clarify that our first results figure (Figure 3\) helps demonstrate this capability. For the index-cued task, we match real behavioural hallmarks from macaque data, specifically the \~10% swap error rate observed experimentally. We then extend to feature-cued networks using swap dependencies that reflect real human psychophysical data (Radmard et al., 2025; Schneegans & Bays, 2017\) in order to make predictions about location-varying experiments. BNS was used to quantify how well our trained networks' outputs matched their target distributions.
>
> Furthermore, in response to comments from multiple reviewers, we have added extensive validation on a multisensory cue combination task (Appendix F), using parameters from an ideal observer model previously fit to human behaviour. We provide quantitative evidence that our method successfully reproduces these complex posteriors across the full stimulus space.
>
> **On RNNs not being the sole option for testable hypotheses:**
>
> We are not sure what is the alternative option the review is suggesting for a testable hypothesis. Whilst we agree that latent variable models constitute a broader class, we emphasise that our work specifically aims to generate hypotheses about neural network mechanisms, not general latent dynamical systems. This necessitates starting from a neural network architecture, with an RNN rather than a feedforward architecture is the appropriate choice given real neural connectivity.
>
> One could alternatively train more abstract but interpretable latent dynamical systems (e.g., Duncker et al., 2019), though this would sacrifice the "cellular-level" interpretability our approach affords. Importantly, the diffusion model-based training procedure we develop here would also be applicable to training such latent dynamical systems to fit rich, multimodal response distributions—representing a valuable direction for future work.
>
> **On biological plausibility:**
>
> We mention biological plausibility for a specific reason: achieving multimodal response distributions required architectural constraints beyond standard rate-based RNNs.
>
> We started with standard architectures employing linear projections followed by point non-linearities (e.g., Yang et al., 2019), but we consistently struggled to achieve multimodal dynamics. A two-layer network at each timestep could produce multimodality, but this is biologically implausible as it implies an intermediary neural population with instantaneous responses (zero time constant). We therefore adopted dendritic tree architectures (Lyo & Savin, 2024), which constrain a deep network's weights to block-diagonal form, producing tree-like connectivity that reflects cortical pyramidal neuron morphology whilst enabling the required distributional flexibility.
>
> We agree that future extensions could incorporate additional constraints such as Dale's law and biologically realistic learning rules. If the reviewer believes this architectural rationale would benefit readers, we would be happy to expand this discussion in the appendix.
>
> **Addressing specific clarity issues:**
>
> * **Line 253 (network output)**: The reviewer's understanding is correct; we have clarified in the revised text (highlighted in blue) that "output" refers specifically to activity in the behavioural subspace **x**.
> * **Lines 208-214**: We have revised this section for clarity
> * **Figure 7A axes**: We have added explicit labels and caption text clarifying that the horizontal and vertical axes of **s\_t** represent the tuning curve activations for probe and report feature dimensions respectively.
>
> With these substantial clarifications and additions (particularly the comprehensive validation on the multisensory task and improved explanation of our experimental progression) we hope the reviewer will consider increasing their score.

---

> > ### Comment · Reviewer_eTxk · 2025-11-27
> > **Response to rebuttal**
> >
> > I thank the authors for the detailed responses.
> > Regarding the dendritic tree - I think this is not a detail for the appendix. You propose a new method that replaces task optimization with distribution matching. In principle, this method can apply for all tasks and all network architectures. If the "standard" RNN does not work with this method, this is a major point to discuss.
> >
> > While real neurons have dendrites, these are often ignored in models. Many details are often ignored - short term synaptic plasticity, excitability dynamics, neuromodulators, axonal conduction delays, and more. It is perfectly valid to state that one such feature (dendrites) happens to be crucial for your method to work. But such a statement should appear in the main text.

---

> > > ### Author Response · Authors · 2025-11-28
> > >
> > > We thank the reviewer for their feedback on this matter. For space purposes, we have made a note in the main text at the first mention of the dendritic architecture, signalling that this architecture choice was partly motivated by the minimum requirements of expressivity, directing the reader to a more detailed discussion in the appendix. We also mention that while a dendritic structure is not necessary for our method, it is a proven (Lyo & Savin, 2024), expressive structure that maintains plausibility in this regard. We again thank the reviewer for their feedback, and we believe this change better motivates our use of the dendritic architecture.

---

### Official Review · Reviewer_LJQh · 2025-11-04

**Soundness:** 4
**Presentation:** 4
**Contribution:** 4
**Rating:** 10
**Confidence:** 3

**Summary:**

In this submission, the authors propose an alternative method to train models of biological visual processing that don't rely on task-optimization. The proposed method aims to synthetically generate a large amount of behavioral data and introduces a diffusion-based method to train RNNs to fit the generated behavioral data, hence automating the development of brain-aligned dynamical system models without a human-in-the-loop. RNN models of a multi-item delayed estimation problem are trained to display 'swap errors' which are characteristic of human behavior on such visual working memory problems, and a representational geometry analysis is shown to highlight how the trained RNN models capture key indicators of visual working memory behavior.

**Strengths:**

- Training biologically plausible dynamical systems models using deep neural networks has been saturated for a while with task-optimized neural networks. The proposed approach in this paper to automate the process of generating ecologically valid synthetic behavioral data and a new paradigm to train RNNs on the above generated data with denoising diffusion is clearly a fresh take that is well distinguished from task-optimized network training.
- While the proposed method is shown to model behavior in a visual working memory task, I believe the general framework proposed here has the potential to scale to, and help interpret other dynamic biological processes in perception and decision making.

**Weaknesses:**

- While task-optimized networks are notoriously difficult to design and train, they provide a direct way to compute representational similarity relative to biological brains on naturalistic stimuli (images, videos, audio, etc). The proposed methods here, however, train RNNs that operate at a different level of abstraction of the input if I understand correctly. This is expected, as the proposed model is one of biological decision making and not necessarily perception. But if one were to apply the proposed method to model perceptual processes, could the authors please comment on how one could bridge this gap in input complexity consumed by their proposed RNN models? Would BNS style synthetic data generation even be feasible for generating coherent, high-dimensional sensory behavioral data?

**Questions:**

NA. Please refer to my review above.

---

> ### Author Response · Authors · 2025-11-23
>
> We are deeply grateful to the reviewer for their exceptionally positive assessment and for recommending our work as a spotlight or oral presentation. We are delighted that the reviewer recognises the novelty and potential impact of our approach.
>
> **On extending to perceptual processes and naturalistic stimuli:**
>
> We thank the reviewer for this thoughtful question about scaling our method to perceptual tasks. We wish to clarify our current approach, though we may have misunderstood the reviewer's concern:
>
> Our feature-cued networks already employ biologically-inspired sensory representations through palimpsest conjunctive tuning curves (Appendix B; Matthey et al., 2015), which are standard in computational neuroscience for modelling early sensory processing. Furthermore, the multisensory cue combination task we added (Appendix F) is explicitly a perceptual/sensory task, modelling how the brain integrates visual and auditory spatial cues.
>
> Regarding naturalistic, high-dimensional stimuli (raw images, video): we envision a version of our approach where pretrained perceptual networks extract features that serve as inputs to our RNNs. We agree that this represents an exciting direction for future work and appreciate the reviewer highlighting this opportunity.
>
> We once again thank the reviewer for their strong endorsement and helpful feedback.

---

> > ### Comment · Area_Chair_uJhR · 2025-11-27
> >
> > Hi Reviewer,
> >
> > The authors have submitted their responses to your reviews. Please take a look and let the authors know if you have any further questions or concerns. Thank you again for your contributions to ICLR!
> >
> > Best regards, AC

---

### Author Response · Authors · 2025-11-23

We sincerely thank all reviewers for their constructive feedback and thoughtful engagement with our work. We are particularly grateful for the enthusiasm expressed about the novelty and potential impact of our approach.

In response to the reviews, we have made substantial improvements to our submission, all of which are highlighted in blue:

**New task domain (Appendix F)**: We present comprehensive results on a multisensory cue combination task, demonstrating that our method successfully captures complex, multimodal posteriors in an entirely different cognitive domain. This addition provides both quantitative validation and establishes the generalisability of our approach beyond VWM.

**Enhanced clarity on network variants**: We have substantially revised the manuscript to explicitly distinguish between index-cued and feature-cued networks. This clarifies our experimental progression: validating our approach against existing neural data (index-cued) before extending to generate testable predictions (feature-cued).

**New neural analyses (Figure 5B)**: We have added linear decoder analyses for index-cued networks following the methodology of Alleman et al. (2024). These analyses provide additional evidence that swap errors in our networks arise during memory retrieval rather than through misbinding during encoding.

**Quantitative metrics for feature-cued networks (Appendix C)**: We now provide quantitative comparisons of plane alignment trajectories across network variants by fitting sigmoid functions and comparing mean squared error against flat-line baselines. These metrics further demonstrate that only networks trained with realistic, distance-dependent swap errors capture the characteristic temporal dynamics observed in biological data.

**Improved clarity throughout**: We have revised language throughout the manuscript, added explicit forward references, corrected errors, and enhanced figure captions to improve accessibility and flow.

We believe these revisions have substantially strengthened the manuscript and hope the reviewers will find them satisfactory.

---

### Author Response · Authors · 2025-12-04

Dear Area Chair,

We are grateful to you for taking on the additional workload during these extraordinary circumstances, and we appreciate the effort required to fairly evaluate submissions given the disruption to the normal review process. We would like to summarise the outcomes of the rebuttal period for your consideration.

## Reviewer engagement
Prior to the data leak, two reviewers explicitly increased their scores in response to our revisions, bringing our overall scores from 10/6/4/4 to 10/6/6/6:
* Reviewer LJQh (10): Recommended our work as a spotlight or oral presentation, recognising the novelty of our approach and its potential to scale to other dynamic biological processes in perception and decision making.
* Reviewer eTxk (6): Raised important points about the role of dendritic architecture in enabling multimodal dynamics. In response, we added discussion of expressivity requirements to both the main text and appendix, clarifying the motivation behind our architectural choices and their connection to cortical pyramidal neuron morphology.
* Reviewer 96uo (raised to 6): Initially raised concerns about qualitative results, limited task domain, and baseline comparisons. After we added the new multisensory cue combination task with quantitative validation, clarified our methodological uniqueness, and provided complete dataset documentation, they increased their score.
* Reviewer YEvN (raised to 6): Raised concerns about dependence on generative models and the need for quantitative validation. After we added the new task domain with Kolmogorov-Smirnov statistics and clarified the role of BNS in our pipeline, they increased their score.


## Changes to submission
All changes are highlighted in blue in the revised manuscript. In response to reviewer feedback, we made the following substantive additions:

* New task domain (Appendix F): We present comprehensive results on a multisensory cue combination task for causal inference, using parameters from an observer model previously fit to human behaviour (Beierholm et al., 2007). This demonstrates that our method successfully captures multimodal behavioural distributions beyond the VWM case. We provide quantitative validation via Kolmogorov-Smirnov statistics comparing ideal and empirical posterior distributions across the full stimulus space.
* New neural analyses (Figure 5B): We added linear decoder analyses for index-cued networks following the methodology of Alleman et al. (2024). These analyses provide additional evidence that swap errors in our networks arise during memory retrieval (delay 2) rather than through misbinding during encoding, matching the conclusions drawn from macaque neural data.
* Quantitative metrics for plane alignment (Appendix C): We now provide quantitative comparisons of plane alignment trajectories across network variants. Specifically, we fit sigmoid functions to alignment dynamics (following Panichello & Buschman, 2021) and compare mean squared error against flat-line baselines. These metrics demonstrate quantitatively that only networks trained with realistic, distance-dependent swap errors capture the characteristic temporal dynamics observed in biological data.
* Enhanced clarity on network variants: We substantially revised the manuscript to explicitly distinguish between index-cued and feature-cued networks, with a clear rationale provided at the beginning of each results subsection. This clarifies our experimental progression: validating our approach against existing neural data (index-cued) before extending to generate testable predictions (feature-cued).
* Architecture expressivity discussion: Following Reviewer eTxk's feedback, we added explicit discussion of our choice of dendritic tree architectures to the main text and appendix, clarifying their role in achieving multimodal dynamics.

We believe these revisions, together with the positive engagement from all reviewers prior to the incident, demonstrate that our work has substantively addressed the concerns raised during the review process.

Thank you again for your consideration.

Best regards,

Submission 14123 Authors

---

### Meta-Review · Area_Chair_aXxe · 2025-12-24

**Summary:**

This paper proposes a diffusion-based procedure to train RNNs to distributions of behavioral data, in particular the behavior of non-human primates engaged in a visual working memory task. The method is shown to possess the ability to match complex target distributions, recover network dynamics that are consistent with recorded neural data (including in terms of neural geometry similarity), and to make novel predictions about the mechanisms of swap errors.

The paper has been praised for its novelty in contrast to the common procedure in theoretical and computation neuroscience to train models to be task-optimal, as it proposes instead to fit the actual behavioral data, including errors (a procedure that is however quite consolidated for instance in model-based RL with fMRI data where fitting trial-by-trial human behavior is common practice, as the authors might want to clarify).

The presentations has been indicated as patchy, with a strong background section, but unclear methodological motivations, and incoherent justifications for some design choices. The rebuttal and the revised version have however addressed these concerns with a more structured presentation, clearer exposition of the rationale and motivations, and more coherent justifications for the design choices.

In addition, the main methodological concerns indicated by the reviews, i.e. the qualitative evaluation of the results and lack of rigorous quantitative data, the narrow score with the inclusion of only one task, and insufficient baseline comparisons have been addressed decisively in the rebuttal and the substantial revisions. These include the modeling of an additional multisensory integration task, more extensive analyses, and statistical comparisons of networks dynamics.

The revised version, with its clearer presentation, expanded methodology, additional task, rigorous data analysis, and strengthened claims of novelty, successfully addressed the main initial concerns and presented a robust contribution to the field of cognitive neuroscience modeling.

**Reviewer Concerns:**

* Addressed in the rebuttal:
  - Requests for missing clarifications on the coding schemes being used in the model have been addressed in the rebuttals and revisions
  - Requests for clarification regarding the design choice of using RNNs have been addressed in the rebuttals and revisions, although that raises questions on the significance of the concept of "biological plausibility" in the context of the paper and more in general
  - Minor reformatting request that would however improve readability have been addressed in the rebuttals and revisions

* Not addressed in the rebuttal:
  - Concerns regarding the limited baseline comparisons and ablations have only been partially addressed in the rebuttals but ultimately authors acknowledged that the provided direct comparisons are limited, justifying that via the novelty of the approach
  - Claims of "automatic discovery" have not been fully justified, as the method still requires significant design choices and human intervention
  - Concerns about the generalizability of the method to more complex tasks requiring for instance continuous outputs (e.g. motor control tasks) have not been fully addressed, as authors acknowledged the limitation, alluding that it could be the object of study of future work

**Reviewer Scores:**

| Reviewer | initial score | predicted final score |
|---:|---:|---:|
| LJQh | 10 | 10 |
| eTxk | 6 | 6 |
| 96uo | 4 | 6 (explicitly said by reviewer) |
| YEvN | 4 | 6 (explicitly said by reviewer) |

---

### Decision · Program_Chairs · 2026-01-26

Accept (Poster)